# Frustrated flexibility in metal-organic frameworks

Roman Pallach [1], Julian Keupp [2], Kai Terlinden[1], Louis Frentzel-Beyme [1], Marvin Kloß [1], Andrea Machalica[1], Julia Kotschy [3], Suresh K. Vasa[3], Philip A. Chater [4], Christian Sternemann [5], Michael T. Wharmby [6], Rasmus Linser [3], Rochus Schmid [2] & Sebastian Henke [1✉]

Stimuli-responsive flexible metal-organic frameworks (MOFs) remain at the forefront of porous materials research due to their enormous potential for various technological applications. Here, we introduce the concept of frustrated flexibility in MOFs, which arises from an incompatibility of intra-framework dispersion forces with the geometrical constraints of the inorganic building units. Controlled by appropriate linker functionalization with dispersion energy donating alkoxy groups, this approach results in a series of MOFs exhibiting a new type of guest- and temperature-responsive structural flexibility characterized by reversible loss and recovery of crystalline order under full retention of framework connectivity and topology. The stimuli-dependent phase change of the frustrated MOFs involves non-correlated deformations of their inorganic building unit, as probed by a combination of global and local structure techniques together with computer simulations. Frustrated flexibility may be a common phenomenon in MOF structures, which are commonly regarded as rigid, and thus may be of crucial importance for the performance of these materials in various applications.

[1] Anorganische Chemie, Fakultät für Chemie und Chemische Biologie, Technische Universität Dortmund, Dortmund, Germany. [2] Computational Materials Chemistry Group, Fakultät für Chemie und Biochemie, Ruhr-Universität Bochum, Bochum, Germany. [3] Physikalische Chemie, Fakultät für Chemie und Chemische Biologie, Technische Universität Dortmund, Dortmund, Germany. [4] Diamond Light Source, Harwell Campus, Didcot, Oxfordshire, UK. [5] Fakultät Physik/DELTA, Technische Universität Dortmund, Dortmund, Germany. [6] Deutsches Elektronen-Synchrotron (DESY), Hamburg, Germany. ✉email: sebastian.henke@tu-dortmund.de

The search, exploration and synthetic control of stimuli-responsive materials, which are able to change their physical properties in response to external triggers, is of highest relevance for future technology and thus continues to draw attention from various fields of materials research; particularly in solid-state, polymer, supramolecular and biological sciences[1–9]. Metal-organic frameworks (MOFs) are a class of porous crystalline materials constructed from inorganic building units, which are interconnected by organic linkers[10,11]. Several MOFs undergo reversible structural transformations in response to external stimuli, such as guest adsorption, changes in temperature, mechanical pressure, light irradiation or electric fields,[12] and thus form an important subclass, termed soft porous crystals or simply flexible MOFs[13–15]. Their structural behaviour renders flexible MOFs promising candidates for applications in the fields of gas storage[16], separations[17], chemical sensing[18,19], controlled drug release[20] or as shock absorbers[21]. One particularly important mode of MOF flexibility is classified as breathing. This term refers to structural transitions driven by guest ad- or desorption, involving significant changes in the crystallographic unit cell volume and the corresponding pore size of the MOF[13,15].

Flexible MOFs switching between two or more well-defined crystalline phases are widely explored and well studied[13,15,22–26]. By contrast, flexible MOFs undergoing reversible transitions between crystalline and non-crystalline phases are only poorly understood[27–32]. Generally, highly disordered or amorphous MOF phases, despite being described more and more frequently in the literature, are rarely investigated in detail, because their disordered nature makes them much more difficult to characterize than their crystalline relatives. Nevertheless, there is growing interest in amorphous MOFs and their importance is being increasingly realized[33–40]. Exciting examples are meltable and glass-forming MOFs, which offer moldability and thus can be processed and shaped in their (supercooled) liquid state—a unique feature which is unachievable for crystalline MOFs[41].

The stimuli-dependent structural flexibility of MOFs typically arises from a delicate balance between enthalpic (e.g. dispersion interactions) and entropic contributions (e.g. vibrational degrees of freedom)[22,24,25]. It has previously been shown that integration of functional groups at the organic linkers of MOFs can drastically change the MOF's energy landscape due to the presence of additional dispersion interactions between these groups and the framework backbone[25,42,43]. In these cases, guest-removal leads to a transition of the flexible MOF from an expanded phase to a contracted phase, in which intra-framework dispersion interactions are maximized. Based on these fundamental observations, we demonstrate here a new kind of stimuli-driven structural responsiveness of functionalized MOFs, which we term frustrated flexibility. The functionalized MOFs feature highly crystalline, cubic framework structures in their solvated state. Upon guest removal, competing forces within the frameworks, namely strong and directional covalent bonding of the organic linker and coordinative ligand-to-metal bonding vs. weaker and non-directional dispersion interactions between neighbouring organic linkers, give rise to degenerate and highly distorted ground states. The distorted MOF phases exhibit the same topology (same connectivity and bonding pattern) as the parent crystals but are devoid of crystalline order.

Our design of frustrated flexibility is based on the following concept. At first, we select an intrinsically rigid and non-responsive MOF structure type, which does not allow for correlated (and thus crystalline-to-crystalline) structural changes involving a hinging or wine rack flexibility mechanism (e.g. breathing transitions). Secondly, the organic linkers of the non-responsive MOF scaffold are decorated with functional groups that act as dispersion energy donors (DEDs) and thus change the

energy landscape of the MOF structure, i.e. the balance of enthalpic contributions (covalent and coordinative bonding as well as dispersion interactions) and entropic contributions (static and dynamic disorder). As a consequence of additional intra-framework dispersion interactions mediated by the DEDs, the previously non-responsive MOF crystals enter a state of frustration once interacting guest molecules are removed from the framework's cavities. In the frustrated state, the MOF is unable to maximize the intra-framework dispersion interactions (i.e. to minimize enthalpy) whilst retaining crystalline order.

For a first proof of this concept, we selected the iconic MOF-5 (chemically $Zn_4O(bdc)_3$, $bdc^{2-}$ = 1,4-benzenedicarboxylate) as a rigid MOF platform[44]. MOF-5 is constructed by the interlinkage of the sixfold connecting $[Zn_4O]^{6+}$ clusters through $bdc^{2-}$ units to a framework of primitive cubic (**pcu**) topology (Fig. 1a). Although there are considerable dynamics suggested to be present in this material, be it the dynamic binding of DMF (N,N-Dimethylformamide) to its metal clusters[45], negative thermal expansion[46,47] or the possibility of chiral induction by guest uptake[48], it is generally considered a structurally rigid MOF[49–51]. MOF-5 always retains its cubic symmetry and does not show any kind of volume change >4%[48] upon guest removal/exchange or gas sorption.

An important signature of the rigidity of MOF-5 is its fairly isotropic Young's modulus, which varies only by a factor of about 1.5 between the least stiff and the stiffest crystallographic direction, as shown by molecular dynamics simulations applying the QuickFF force field[52]. Flexible MOFs typically feature orders of magnitude differences of their Young's moduli in different crystallographic directions[53]. In view of its topology, the non-responsiveness of MOF-5 is surprising, since several other MOFs exhibiting the **pcu** topology are known to feature remarkable guest- and temperature-responsive phase transitions involving extremely large volume changes[25,42,54,55]. The inherent rigidity of MOF-5 originates from the geometric features of its $T_d$ symmetric $[Zn_4O(O_2C)_6]$ nodes. The nodes do not allow for a cooperative hinging of its $bdc^{2-}$ linkers in opposite directions, for instance along the body diagonal of the cubic cavity (the $\langle 111 \rangle$ direction, Fig. 1). Thus, large-magnitude volume changes of the crystal, as in the prominent breathing phase transitions of conventional flexible MOFs, are not feasible for MOF-5[49], since they would require major distortions of its inorganic nodes or even reversible bond breaking.

In the following, we demonstrate the phenomenon of frustrated flexibility in a series of functionalized MOF-5 derivatives, which feature additional intra-framework dispersion forces competing against the strong and directional coordination bonds responsible for structural rigidity. Via chemical functionalization of the $bdc^{2-}$ units with additional alkoxy groups as DEDs, remarkable guest- and temperature-responsive structural flexibility is initiated in MOF-5. By tuning the steric bulk of the DEDs at the backbone of the MOFs, the mode and the degree of distortion of the guest-free phase can be modulated systematically from correlated and crystalline distortions (up to 3% volume contraction) to random and non-crystalline distortions (up to 17% volume contraction). Importantly, the non-crystalline phases can be reconverted to the crystalline phase via guest adsorption or simply by heating the material to elevated temperature. This endows frustratedly flexible MOFs with unprecedented physical properties, such as continuous non-crystalline-to-crystalline transitions driven by entropy rather than enthalpy.

## Results

**MOF synthesis**. Seven functionalized MOF-5 derivatives of the general chemical composition $Zn_4O(CX\text{-}bdc)_3$ ($CX\text{-}bdc^{2-}$ = 2,5-dialkoxy-1,4-benzenedicarboxylate with X = 2–8: number of

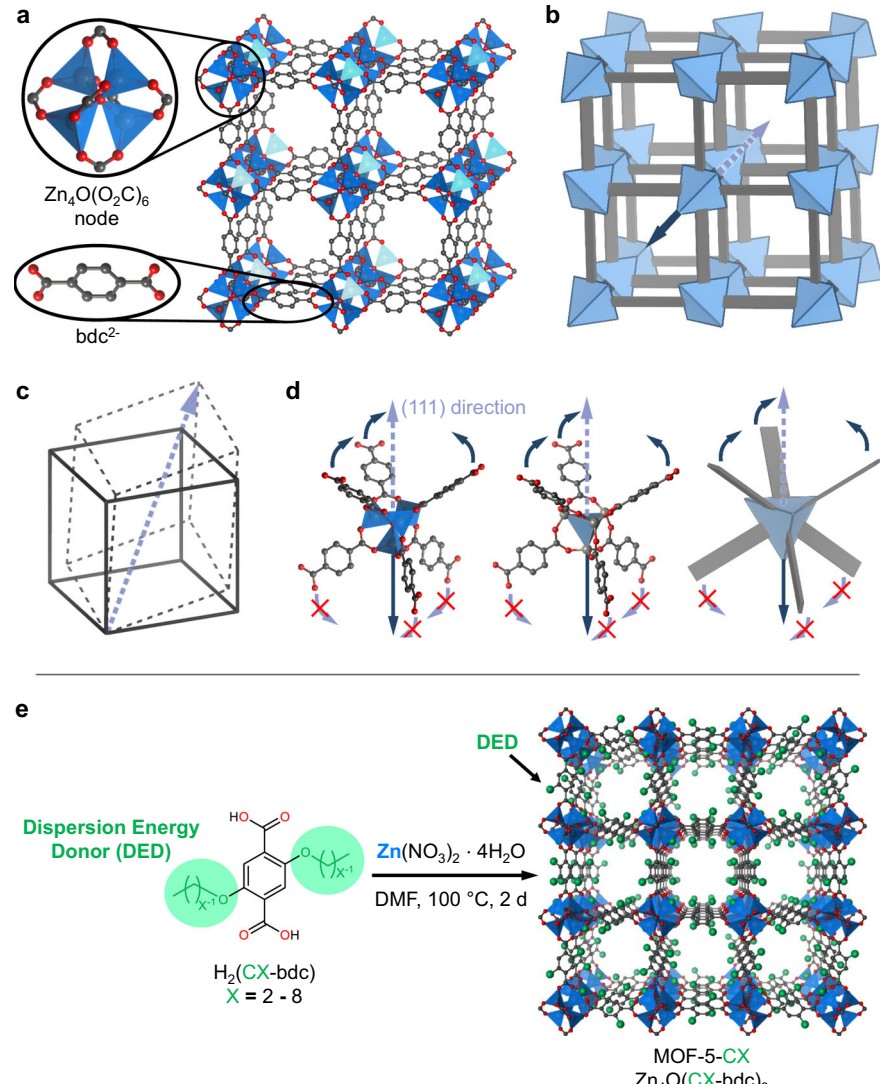

**Fig. 1 Structure and geometrical constraints of the MOF-5 structure type. a** Crystal structure of MOF-5 (CCDC deposition code 256965). Zn, O and C atoms are shown in blue, red and grey, respectively. H atoms are omitted. **b** Simplified building block representation of the structure of MOF-5. **c** Rhombohedral distortion of a cube along its body diagonal. **d** Abstraction path of the actual chemical structure towards the respective geometries of its building units as shown in the simplified building block representation (**b**). Arrows symbolize the required movements of these building units (that is, $[Zn_4O]^{6+}$ and $bdc^{2-}$) in case of the rhombohedral distortion. **e** Schematic for the preparation of alkoxy-functionalized MOF-5-CX derivatives. Dispersion energy donors (DEDs) are marked as green spheres. The atom colour code is the same as in (**a**).

carbon atoms in the alkyl chain, Fig. 1e), bearing linear alkoxy groups of variable length at the organic $bdc^{2-}$ linker, were prepared as transparent cubic crystals (size ~100–500 μm), denoted as-MOF-5-CX. Repeated solvent exchange with $CH_2Cl_2$ (three times) followed by careful drying under dynamic vacuum (~$10^{-4}$ kPa) at 100 °C for 24 h gives the guest-free, dried materials dry-MOF-5-CX.

**Guest-responsive structural behaviour.** Powder and single crystal X-ray diffraction (PXRD and SCXRD) data of the as-synthesized (as) materials as-MOF-5-C2 to as-MOF-5-C7 confirm their structural analogy to prototypical MOF-5 (Fig. 2a, Supplementary Methods 2 and 3). The material as-MOF-5-C8 possesses a different structure, as indicated by characteristic peak splittings in its PXRD pattern. Indeed, SCXRD verifies that as-MOF-5-C8 is a rhombohedrally distorted phase of MOF-5 with angles of 87.0° and 93.0° between

neighbouring inorganic nodes (space group $R\bar{3}$) instead of 90° in the cubic parent structure.

Due to the geometric constraints of the $[Zn_4O(O_2C)_6]$ nodes, the rhombohedral distortion of as-MOF-5-C8 involves a slight deformation of the $[Zn_4O]^{6+}$ tetrahedron along the rhombohedral elongation axis (the body diagonal of the framework cavity), giving rise to two different kinds of Zn–O distances, three longer ones of 1.977(1) Å and one shorter one of 1.925(1) Å (Supplementary Figs. 3.1 and 3.2). For reference, the Zn–O bond length in MOF-5 amounts to 1.935(2) Å and lies in between the Zn–O bond lengths of the rhombohedral as-MOF-5-C8. Remarkably, the alkoxy DEDs connected to the linker could be well resolved in the electron density map of as-MOF-5-C8. The DEDs are located in close contact to the aromatic ring of a neighbouring $bdc^{2-}$ linker (~3.0–3.5 Å $C_{aryl}$-$H_{DED}$ distance), giving clear evidence that dispersion interactions between these groups provide the driving force for the observed rhombohedral distortion.

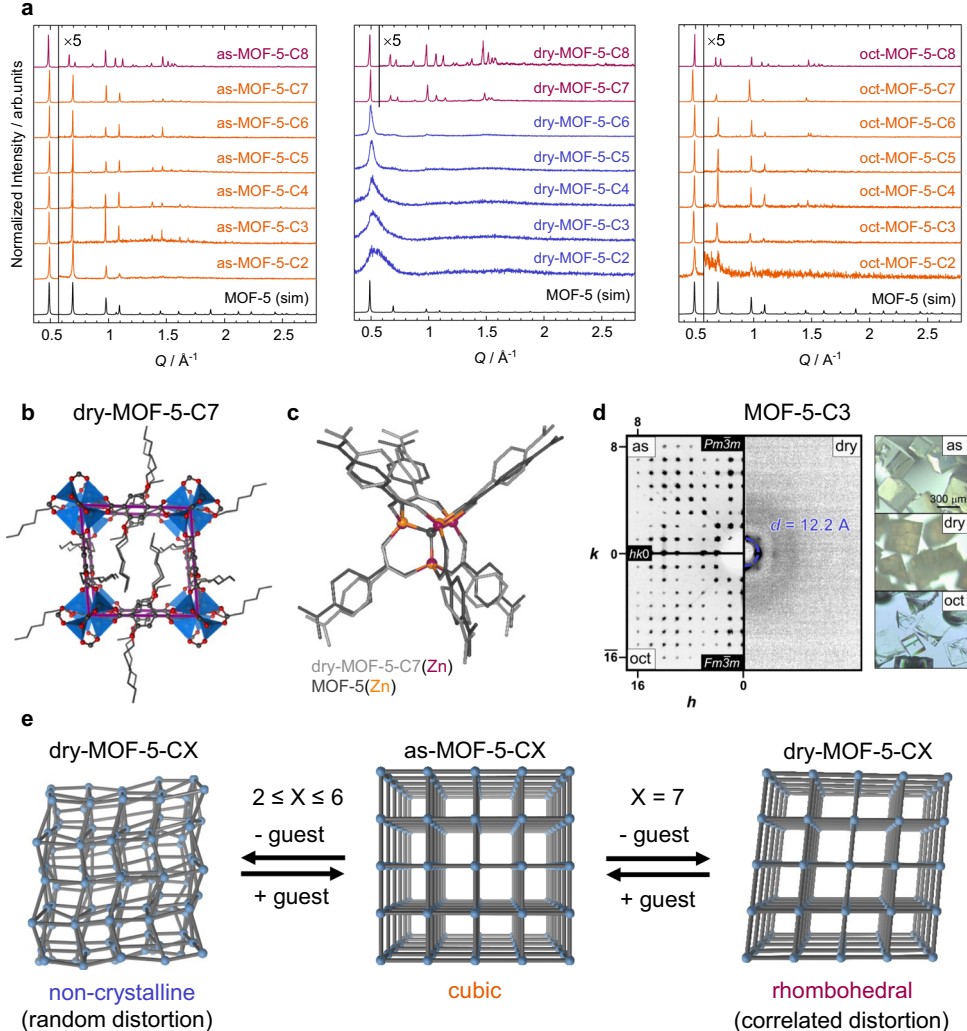

**Fig. 2 Guest-responsive structural behaviour of MOF-5-CX. a** PXRD data of the MOF-5-CX in their as (as-synthesized), dry and oct (*n*-octane resolvated) states in comparison to the simulated diffraction pattern of MOF-5 (CCDC deposition code 256965). Cubic, rhombohedral and non-crystalline states are highlighted in orange, purple or blue colour, respectively. **b** Rietveld refined crystal structure of rhombohedrally distorted dry-MOF-C7 (space group $R\bar{3}$, unit cell edges coloured in purple). **c** Overlay of the structures of regular MOF-5 (grey, zinc cations in orange) and dry-MOF-5-C7 (light grey, zinc cations in purple) with emphasis on the relative distortion of the $[Zn_4O(O_2C)_6]$ building unit. **d** SCXRD data showing the loss and recovery of the (single-) crystalline order from as- to dry- to oct-MOF-5-C3 and microscopy images illustrating the visual impact of the structural transformations. **e** Graphical representation illustrating the guest-responsive structural behaviour of the MOF-5-CX series.

Drying of the MOF-5-CX derivatives gives rise to an unexpected structural richness as a function of the carbon chain length of the DEDs. The PXRD pattern of dry-MOF-5-C8 remains largely the same, whilst dry-MOF-5-C7 likewise features a rhombohedrally distorted structure (Fig. 2a). A Rietveld refinement against a high-resolution PXRD pattern of dry-MOF-5-C7 based on a modified crystallographic model of as-MOF-5-C8, yields a structural model with rhombohedral angles of 84.5° and 95.5° between neighbouring $[Zn_4O(O_2C)_6]$ nodes (Fig. 2c), thus demonstrating a volume contraction of the framework of ~3% upon guest removal (Supplementary Methods 2.2). The cubic-to-rhombohedral transition of dry-MOF-5-C7 can be reversed by adsorption of either DMF vapour or liquid *n*-octane (Fig. 2a, Supplementary Fig. 2.8).

A drastically different behaviour is observed upon guest removal from the MOF-5-CX derivatives bearing shorter DEDs at the organic linker. Dry-MOF-5-C2 to dry-MOF-5-C6 lose long-range order and transform to non-crystalline states, displaying only broad diffuse scattering in their PXRD patterns.

The first scattering peak (FSP) is roughly at the same scattering angle of the strong 100 reflection present in the PXRD patterns of the cubic as-synthesized compounds (Fig. 2a). Following the sequence from dry-MOF-5-C6 to dry-MOF-5-C2, the FSP decreases in absolute peak intensity, increases in peak width and asymmetry, and shifts towards higher scattering angles. Peak fits of the FSP using split pseudo-Voigt or PearsonVII profiles yield peak maxima ranging from $Q_{FSP} = 0.496\,\text{Å}^{-1}$ for dry-MOF-5-C6 to $Q_{FSP} = 0.518\,\text{Å}^{-1}$ for dry-MOF-5-C2. These maxima correspond to real-space distances from $d_{FSP} = 12.66\,\text{Å}$ (dry-MOF-5-C6) to $d_{FSP} = 12.12\,\text{Å}$ (dry-MOF-5-C2) (Supplementary Table 2.3). Furthermore, the increase of the FWHM from $0.02\,\text{Å}^{-1}$ (dry-MOF-5-C6) to $0.17\,\text{Å}^{-1}$ (dry-MOF-5-C2) suggests a drastic increase in strain and/or a decrease in domain size with decreasing alkoxy chain length. Since the position of the FSP is characteristic for the average distance between two neighbouring $[Zn_4O]^{6+}$ clusters in the solid, its shift to higher Q demonstrates a contraction of the frameworks concomitantly to the loss of long-range order. Calculation of $(d_{FSP})^3$ allows for an

approximation of the mean volume occupied by a single formula unit of the non-crystalline dry-MOF-5-CX frameworks (Supplementary Table 2.4). Comparing this to the unit cell volume of the parent crystalline as-MOF-5-CX phases yields an estimated volumetric contraction from −6% for dry-MOF-5-C6 up to −17% for dry-MOF-5-C2. Macroscopically, with the loss of crystalline order upon desolvation, the formerly transparent crystals turn dull and cracked, due to the strain built up within the material (Fig. 2d).

Importantly, sharp Bragg reflections, characteristic for the cubic MOF-5 structure type, re-emerge in the PXRD patterns of dry-MOF-5-CX (X = 2–6) after adsorption of DMF or n-octane (Fig. 2a, Supplementary Figs. 2.3–2.7). The fact that this phenomenon occurs with n-octane (in which the MOFs' building blocks are not soluble) rules out solvent coordination or framework reconstruction as a potential cause of this effect. Most impressively, the crystalline-to-amorphous-to-crystalline transition after DMF removal and subsequent adsorption of n-octane at 65 °C could even be followed by SCXRD of representative samples of MOF-5-C3 and MOF-5-C6. Reconstructed reciprocal space sections of as-MOF-5-C3 and oct-MOF-5-C3 demonstrate that crystalline order is fully regained at the single-crystal level (crystal size > 300 μm) after immersion of dry-MOF-5-C3 in n-octane (Fig. 2d). Moreover, the structural retransformation can be tracked by the naked eye from a clearing up of the formerly opaque regions of the MOF crystals. Remarkably, the mosaicity of the crystals does not increase after the guest-mediated crystalline-to-amorphous-to-crystalline transition (Supplementary Table 3.4). This indicates that the loss of crystallinity in dry-MOF-5-CX occurs mainly through a strain-based process, rather than cracking or bond breaking.

Ultimately, all the non-crystalline dry-MOF-5-CX materials clearly recover their crystalline cubic phases when either polar or nonpolar guest molecules are reintroduced. This further demonstrates that the amorphization process upon guest-removal cannot originate from an irreversible collapse of the framework via bond dissociation but must be mediated by intra-framework dispersion interactions that are strong enough to result in a non-correlated volumetric contraction under random distortion of the framework (Fig. 2e).

**Local structure.** In order to get further insights into the structural changes during the unprecedented crystalline-to-amorphous-to-crystalline transition of these MOF-5-CX materials, their local structure was studied via solid-state cross polarisation (CP) $^{13}$C magic angle spinning nuclear magnetic resonance (MAS NMR) spectroscopy, X-ray total scattering and infrared (IR) spectroscopy. CP $^{13}$C MAS NMR spectra reveal a generally heterogeneous local structure in case of non-crystalline dry-MOF-5-CX with X = 2–5, as illustrated by the smoothly broadened signals of the carbon atoms belonging to the organic backbone of the MOFs (Fig. 3a). Heterogeneity in dry-MOF-5-C6 is apparent to a lesser extent, with defined individual features from fewer underlying configurations becoming discernible. The rhombohedral dry-MOF-5-C7 and dry-MOF-5-C8 feature sharp resonances for all carbon atoms. Apart from corroborating the above XRD data, these observations prove the overall contribution of the various MOF constituents to the amorphization at the atomic level. It is also important to note that all CP $^{13}$C MAS NMR spectra establish neither breakage nor loss of the carboxylate-to-zinc coordination, even for the strongly distorted dry-MOF-5-C2 and dry-MOF-5-C3 materials, as non-coordinating carboxylate groups are expected at a chemical shift of 180–185 ppm[56].

High-resolution X-ray total scattering data for all as-MOF-5-CX and dry-MOF-5-CX derivatives were recorded on beamline

I15-1 at Diamond Light Source and converted into the corresponding real-space X-ray pair distribution functions (XPDFs) in the form of $D(r)$. For MOF-5-C7 we find a close resemblance of the XPDFs of the cubic (as) and rhombohedral (dry) phases up to around 15 Å, beyond which peaks originating from Zn–Zn distances across the cubic face diagonal show a characteristic change in their positions (Supplementary Fig. 5.9). As expected for MOF-5-CX materials bearing shorter alkoxy chains (MOF-5-C2 to MOF-5-C6), a major difference between the XPDFs of cubic (as) and non-crystalline (dry) materials is found in the entire loss of correlations above $r$ ~13.5 Å for dry-MOF-5-C2 up to $r$ ~28.3 Å for dry-MOF-5-C6 (Fig. 3b and 3c, Supplementary Methods 5). The distance of 13.5 Å mainly corresponds to Zn–Zn correlations between adjacent [Zn$_4$O(O$_2$C)$_6$] nodes (i.e. within one cavity). The distance of 28.3 Å is in the range of two MOF cells (the term "cell", which we define here, refers to the volume of a single Zn$_4$O(CX-bdc)$_3$ repeating unit of the MOF and not to a crystallographic unit cell), thus indicating the presence of very small, ordered domains of a size of about 2 × 2 × 2 MOF cells in dry-MOF-5-C6.

In the region below $r = 4$ Å, the XPDFs of all dry-MOF-5-CX materials feature three characteristic peaks located at 1.4, 2.0 (peak A) and 3.1 Å (peak B; Fig. 3b). These pair correlations are ascribed to nearest neighbour distances of C–C/C–O, Zn–O and Zn–Zn (with minor contributions from Zn–C and further Zn–O distances to the latter). Hence, these short-range distances are characteristic for the geometry of the [Zn$_4$O(O$_2$C)$_6$] node. Guest-removal and the corresponding structural distortions induce a shift of the maximum of the Zn–Zn correlation (peak B in Fig. 3b and 3c) to higher distances, while the Zn–O correlation (peak A) remains largely unchanged. This, in combination with a pronounced broadening of peak B towards higher $r$, clearly indicates drastic distortions of the [Zn$_4$O(O$_2$C)$_6$] nodes in the non-crystalline frameworks. Hence, we deduce that the loss of long-range order in these materials arises from non-correlated random distortions of these inorganic nodes, in contrast to MOF-5-C7 (and MOF-5-C8), where the [Zn$_4$O(O$_2$C)$_6$] nodes undergo distortions in a correlated (and thus crystalline) fashion. In accordance with the PXRD data and illustrated by the peak width ($\Delta r$) of peak B at $D(r) = 0$, the deformation of the nodes increases with decreasing size of the DEDs (dry-MOF-5-C6: $\Delta r = 0.54$ Å; dry-MOF-5-C2: $\Delta r = 0.74$ Å).

A similar picture can be drawn from IR spectroscopic data by analysis of the characteristic stretching modes $\nu_{as}$(Zn-O1) and $\nu_s$(Zn-O2) of the [Zn$_4$O(O$_2$C)$_6$] nodes at 517 and 576 cm$^{-1}$, as well as $\nu_{as}$(COO) of the coordinating carboxylates at 1571–1601 cm$^{-1}$ (Fig. 3d)[57,58]. The rhombohedrally distorted dry-MOF-5-C7 and dry-MOF-5-C8 display sharp and well-defined bands of the $\nu_{as}$(Zn-O1) and $\nu_s$(Zn-O2) vibrations, whereas the non-crystalline dry-MOF-5-CX materials feature only a broad band of increasing width when following the series from dry-MOF-5-C6 to dry-MOF-5-C2. This establishes a reduction of symmetry of the [Zn$_4$O(O$_2$C)$_6$] nodes and might further indicate the presence of differently deformed clusters in the non-crystalline solids. In line with the XPDF data, the increasing width of the vibrational bands suggests a successively increasing distortion of the nodes from dry-MOF-5-C6 to dry-MOF-5-C2. By contrast, resolvated (and again crystalline) oct-MOF-5-CX display sharp bands of the $\nu_{as}$(Zn-O1) and $\nu_s$(Zn-O2) vibrations akin to those of perfectly crystalline MOF-5 (Supplementary Fig. 6.8). Hence, the $T_d$ symmetry of the [Zn$_4$O(O$_2$C)$_6$] building unit is fully restored upon guest readsorption. The $\nu_{as}$(COO) bands of dry-MOF-5-CX show a similar broadening with FWHM values spread from 47 cm$^{-1}$ (dry-MOF-5-C6) to 91 cm$^{-1}$ (dry-MOF-5-C2) as well as a simultaneous shift of the band maxima from 1597 cm$^{-1}$ (dry-MOF-5-C6) to 1571 cm$^{-1}$ (dry-MOF-5-C2).

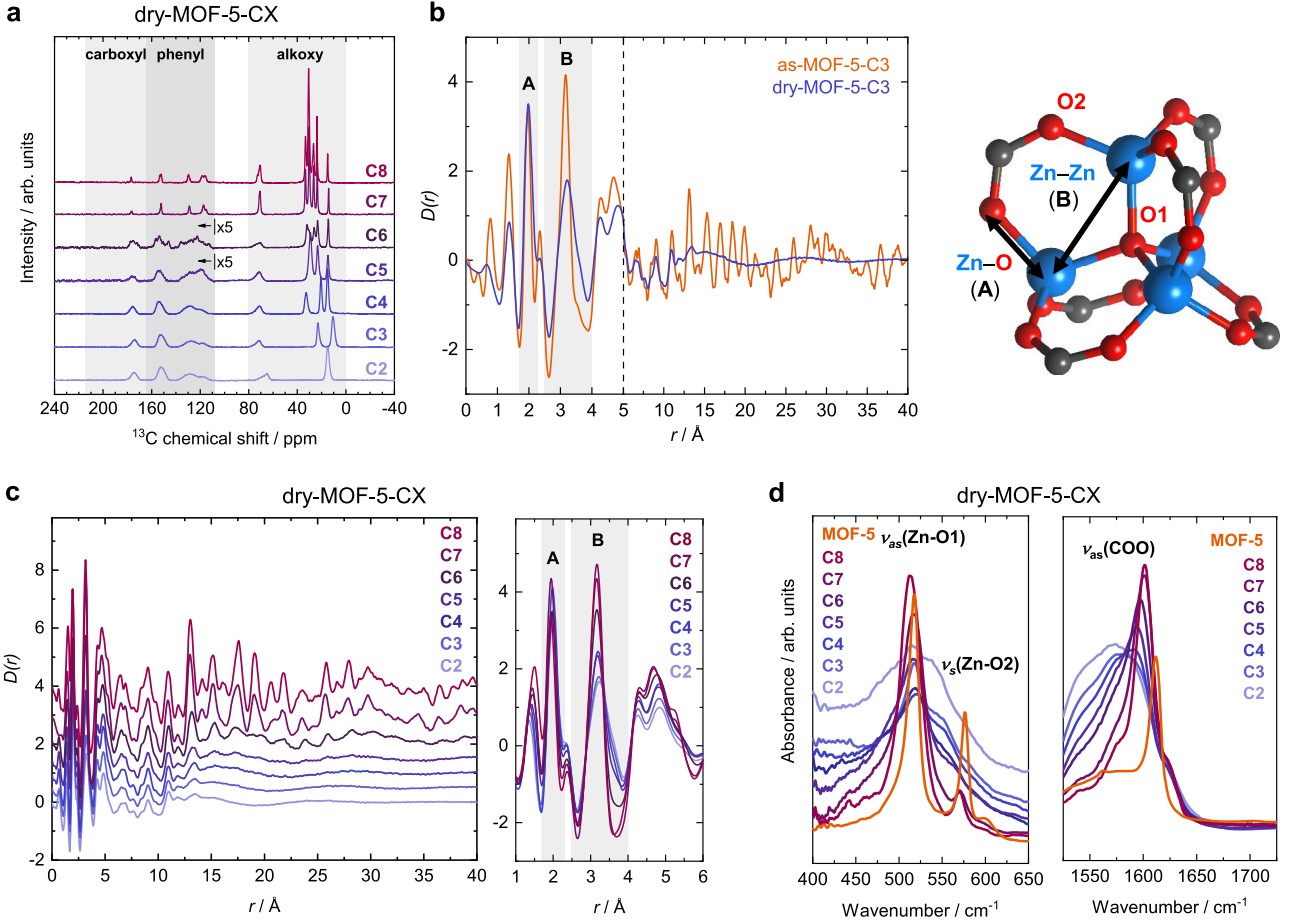

**Fig. 3 Insights into the local structure by solid state NMR, XPDF and IR spectroscopy. a** CP $^{13}$C MAS NMR spectra of the dry-MOF-5-CX with highlighted spectral regions corresponding to carboxylate, phenyl or alkyl carbon atoms. **b** XPDFs of as- and dry-MOF-5-C3 and a representation of the [$Zn_4O(O_2C)_6$] node with distances important for the discussion marked. **c** Overlay of the XPDFs of dry-MOF-5-CX materials with emphasis on the change of the correlation length (left, XPDFs vertically offset) and the low-$r$ region (right). **d** Excerpts from the IR spectral regions showing the $\nu_{as}$(Zn-O1) and $\nu_s$(Zn-O2) and the $\nu_{as}$(COO) vibrational bands of the dry-MOF-5-CX materials.

These findings further demonstrate changes in carboxylate-to-zinc coordination bonds due to the deformation of the [$Zn_4O(O_2C)_6$] nodes[58,59].

**Molecular dynamics simulations.** With the aim to achieve a fully atomistic understanding of the phenomenon of frustrated flexibility, we employed MD simulations using the first principles parameterized force field MOF-FF[60], which was recently extended for pillared-layer-MOFs with alkoxy side chains.[61] To keep these simulations numerically tractable, we have to rely on periodic boundary conditions (PBC) and moderately sized supercells, limiting the formation of long-range disorder. In the smaller 2 × 2 × 2 simulation cells, we computed the thermodynamic properties for the entire series of guest-free MOF-5-CX materials (Supplementary Methods 11). In addition, we performed simulations using a larger 8 × 8 × 8 supercell for MOF-5-C3 and -C6. Since it was recently shown that the linker- and side-chain configuration directly affects the simulation results,[61,62] we chose to derive all initial structural models from the same as-MOF-5-C8 single-crystal structure by stepwise truncation.

The free-energy profiles for the smaller simulation cells were calculated by thermodynamic integration of the pressure vs. volume curves $p(V)$, which were determined in the NV($\sigma_a = 0$)T ensemble.[63] Except for MOF-5-C8, all systems exhibit a cubic structure at ambient conditions. MOF-5-C2 to -C5 also feature a metastable contracted state with a shallow barrier connecting

the minima, indicating a possible deformation. The driving force is clearly the internal energy mediated by the DEDs (Fig. 4b), because for all investigated systems the global internal energy minimum is at a reduced volume, corresponding to a contracted and distorted phase. MOF-5-C8 is special here, because it is the only system with no apparent minimum for the cubic phase, and exhibits—as in the experiment—a rhombohedral structure at ambient conditions. Note that the computed volumes for the contracted local free-energy minima are smaller than the volumes of the dry-MOF-5-CX phases estimated from experimental PXRD data (Fig. 4c). This is presumably a result of the too small simulation box size and insufficient sampling of linker configurations. The results, however, clearly indicate the propensity of all systems for a deformation towards a contracted phase, stabilized by the DEDs.

Interestingly, for the 8 × 8 × 8 simulation boxes of MOF-5-C3 and C6, we indeed observe a tendency for amorphization, however, only upon triggering the phase transformation by increased pressure, since the timescale for this process is out of reach for an unbiased simulation. The crystalline-to-amorphous transition occurs at a moderate pressure of 0.15 GPa and 0.2 GPa for MOF-5-C3 and -C6, respectively (Fig. 4f). The amorphization is obvious from the side view of the simulation box before and after the phase transformation (Fig. 4d, Supplementary Figs. 11.9 and 11.10). Both materials change from a homogenous and narrow cell volume distribution to a very broad one. Here the cell

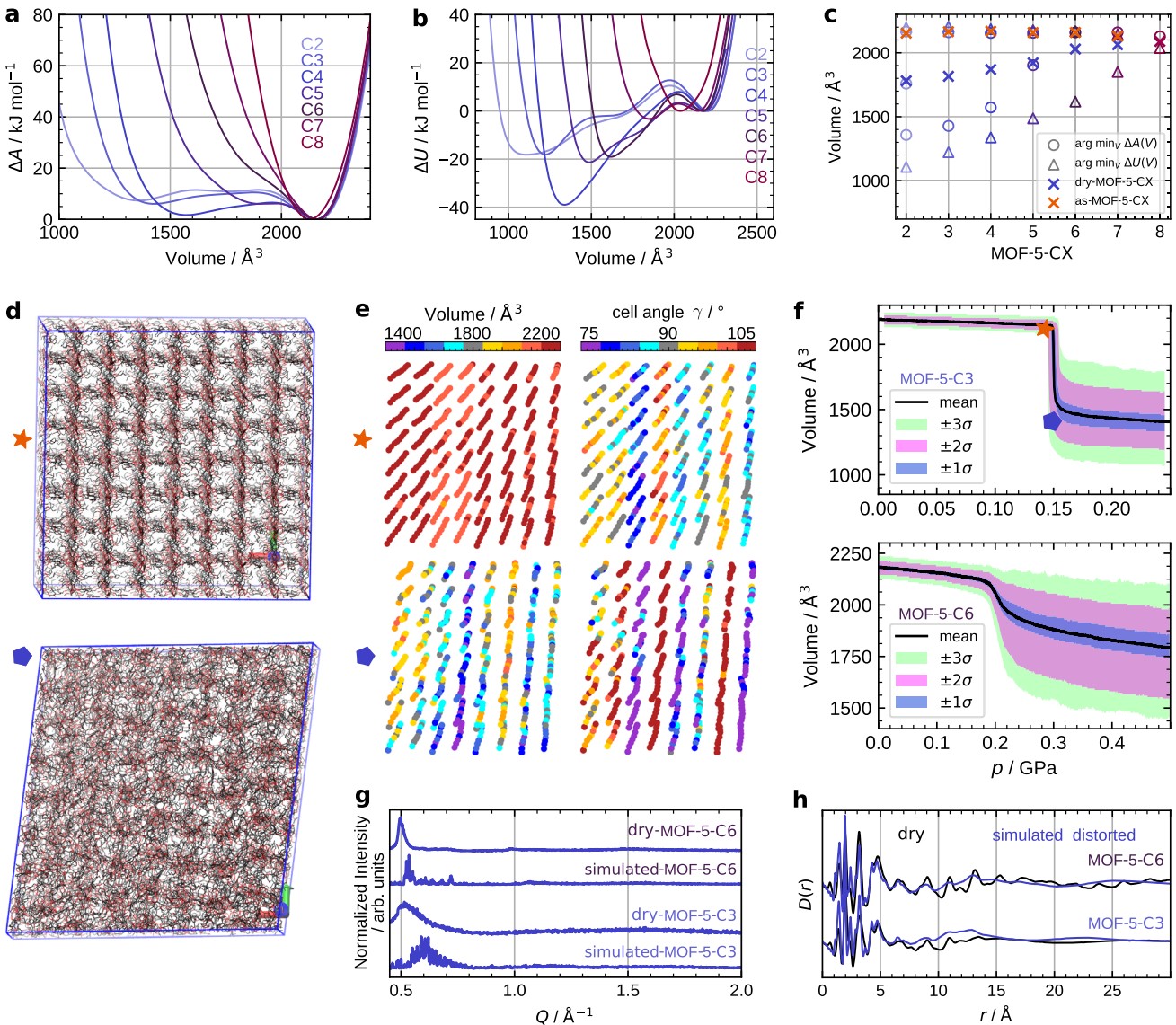

**Fig. 4 Molecular dynamics simulation results. a** and **b** Free energy $\Delta A$ and inner energy $\Delta U$ profiles with respect to volume; **c** Positions of the local energy minima of $\Delta A$ (circles) and $\Delta U$ (triangles) in comparison to the values estimated from experimental PXRD data (crosses). **d** Side view of MOF-5-C3 (large simulation box) before (red star) and after (blue pentagon) the pressure-induced phase transformation. **e** Individual cell volumes and angles $\gamma$ of MOF-5-C3 at these points. **f** Distribution of individual cell volumes for MOF-5-C3 (top) and -C6 (bottom) with respect to pressure, the labels indicate where the snapshots for MOF-5-C3 were taken. **g** and **h** Simulated vs. measured PXRD patterns and XPDFs, respectively, for dry-MOF-5-C3 and -C6.

volume is defined as the volume taken by a single $Zn_4O(CX\text{-}bdc)_3$ formula unit. This is also apparent in the spatial distribution of the individual cell volumes shown for MOF-5-C3 in Fig. 4e. After the phase transformation these volumes are only weakly correlated between neighbouring cells with a correlation length smaller than two cell edge lengths, and thus inhomogeneously distributed throughout the simulation box. However, a somewhat different picture is observed for the cell angles, as indicated by the local angle $\gamma$ for each cell in Fig. 4e. The different cell angle types ($\sphericalangle xy$, $\sphericalangle xz$ and $\gamma = \sphericalangle yz$) form layers along the three spatial directions. These layers appear in different thickness and are more disordered compared to the small cell simulations. A similar correlation has been observed in more pronounced form also for the pillared-layer-type MOFs.[61,64,65]

In good agreement with the experimental data, the hetero-geneous structures of the contracted phases are characterized by a broad distribution of reflections in the simulated PXRD patterns, as well as loss of long-range order beyond 15 Å in the simulated

XPDFs (Fig. 4g and h). Note that the elevated pressure in the simulations naturally results in a shift of the PXRD peaks towards higher $Q$ values as compared to the experimental data. Despite that, the overall shape of the computed PXRD pattern and XPDFs match well with the experimental results. Note that in the simulation, both systems slowly reform the cubic phase upon pressure release (Supplementary Figs. 11.9 and 11.10), which is presumably due to the artificial constraint of PBC and the insufficient sampling time for the DEDs to properly arrange in the reduced pores. However, for MOF-5-C3 we still observe a single distorted layer after the pressure is released. Even though the simulations do not quantitatively match the experimental findings, the results clearly corroborate that a framework-connectivity preserving loss of long-range order driven by intra-framework dispersion forces is plausible. Even at the quite small simulation box dimension of about 10 nm, a contraction of MOF-5-C3 and MOF-5-C6 results in non-correlated structural distortions.

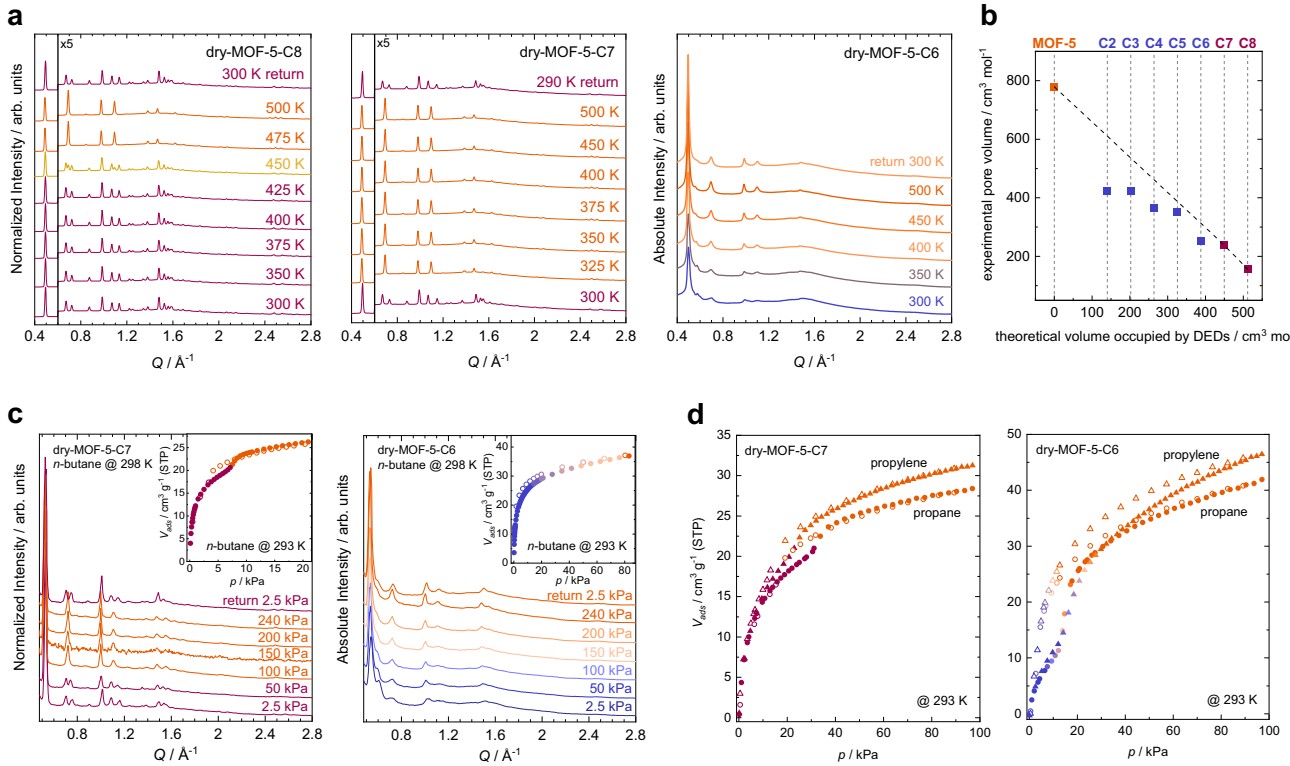

**Fig. 5 Thermal and gas sorption properties of MOF-5-CX. a** Variable temperature (VT-)PXRD data of dry-MOF-5-C8 to dry-MOF-5-C6. **b** Plot of the experimental pore volumes ($CO_2$) vs. the theoretical volume occupied by DEDs in dry-MOF-5-CX. The dashed line signifies the expected pore volumes obtained by linear interpolation from MOF-5 to dry-MOF-5-C8. **c** In situ PXRD data during *n*-butane sorption and associated *n*-butane sorption isotherms (inlet) of dry-MOF-5-C7 and dry-MOF-5-C6. **d** Propane and propylene isotherms of dry-MOF-5-C7 and dry-MOF-5-C6 recorded at 293 K. Adsorption and desorption branches are shown as closed and open symbols, respectively.

**Thermal responsivity**. The dispersion interactions introduced by the DEDs are apparently strong enough to outbalance the energetic cost for the deformation of the nodes, so that the contracted dry-MOF-5-CX materials (either non-crystalline or rhombohedral) are enthalpically favoured in the guest-free state (Fig. 2e). Variable temperature (VT-)PXRD data reveal that vibrational entropy can outbalance the enthalpic stabilization of the contracted phases, resulting in a return to the expanded cubic phases at elevated temperatures. Dry-MOF-5-C8 and dry-MOF-5-C7 undergo fully reversible rhombohedral-to-cubic phase transitions within temperature regimes of 425–450 K and 300–325 K, involving small volume changes of +1.5% and +2.0% (Fig. 5a). The phase transition temperatures ($T_{rh\rightarrow c}$) of the endothermic rhombohedral-to-cubic transitions were determined by DSC as 435 K and 323 K (onset temperature of the calorimetric peak) for dry-MOF-5-C8 and dry-MOF-5-C7, respectively. Both the $T_{rh\rightarrow c}$ and the associated enthalpy change ($\Delta H_{rh\rightarrow c}$) are higher for dry-MOF-5-C8 compared to dry-MOF-5-C7, indicating stronger intra-framework dispersion interactions of the material with the longer alkyl chains (Supplementary Table 7.2).

In contrast, the non-crystalline dry-MOF-5-C6 shows a very different structural behaviour upon heating. The broad scattering peaks in its PXRD pattern constantly get sharper and increase in intensity when the material is heated from 300 to 500 K (Fig. 5a). At 500 K, fairly sharp reflections, which can be indexed in the cubic unit cell of the crystalline derivative as-MOF-5-C6 are evident. Over the range from 300 to 500 K the FSP at $Q \sim 0.5 \text{ Å}^{-1}$ continuously shifts to lower $Q$ values, its integrated intensity increases by 72% and the peak half width decreases by almost a factor of 3 (Supplementary Fig. 8.5, Supplementary Table 8.1). Consequently, dry-MOF-5-C6 shows a continuous transformation

from the non-crystalline to the cubic phase upon heating. In the absence of a latent heat in DSC experiments (Supplementary Fig. 7.5) this behaviour is interpreted as a second-order phase transition, i.e. a continuous structural change of the randomly distorted non-crystalline framework towards the crystalline cubic phase. Interestingly, the reverse transformation from the crystalline to the non-crystalline state of dry-MOF-5-C6 does not instantaneously occur upon cooling to room temperature. Instead, it takes place in a slow and gradual fashion, signifying a slow relaxation to the enthalpically favoured non-crystalline state (Supplementary Fig. 8.6).

The thermal behaviour of dry-MOF-5-C6 is counterintuitive and inverse to the thermal behaviour of conventional materials. Materials typically lose crystallinity with rising temperature, due to entropy-driven defect formation and eventually amorphization or melting. For dry-MOF-5-C6 we observe the opposite behaviour. Due to its frustrated framework structure, the contracted, non-crystalline phase is the enthalpically favoured ground state at room temperature. Increasing the temperature results in a population of vibrational degrees of freedom, which counteract the intra-framework dispersion interactions and thus allow the structure to relax to the expanded crystalline phase. Thus, in the case of dry-MOF-5-C6, the crystalline high-temperature phase must have a higher entropy than the non-crystalline distorted phase. In other words, the configurational entropy of the non-crystalline phase is outbalanced by the vibrational entropy of the crystalline phase at high temperature.

Importantly, the thermal behaviour of dry-MOF-5-C5 and dry-MOF-5-C4 is closely related to that of dry-MOF-5-C6, even though the sharpening of the reflections with temperature gets less prominent with decreasing X (Supplementary Figs. 8.3 and

8.4). For the more heavily distorted dry-MOF-5-C3 to dry-MOF-5-C2 we could not observe a structural response towards the elevated temperatures applied here. Hence, it may be that the temperatures required for the transition from the non-crystalline to the cubic phase are beyond the decomposition temperatures ($T_d$) of the materials ($T_d \sim 605$ K).

**Gas sorption properties**. The porosity of the dry-MOF-5-CX materials was analyzed by $N_2$ (77 K), $CO_2$ (195 K) and $n$-butane (293 K) gas sorption experiments. Only dry-MOF-5-C2 and dry-MOF-5-C3 are able to adsorb $N_2$ at 77 K (capacity of ~200–250 cm$^3$ g$^{-1}$ at saturation), whereas all other derivatives only show $N_2$ adsorption on the external surface (Supplementary Figs. 9.1 and 9.2). However, all dry-MOF-5-CX derivatives readily adsorb $CO_2$ at 195 K, featuring Type 1 isotherms with constantly decreasing quantities of adsorbed gas from dry-MOF-5-C2 to dry-MOF-5-C8. The corresponding BET surface areas range from 1167 m$^2$ g$^{-1}$ (X = 2) to 159 m$^2$ g$^{-1}$ (X = 8). Considering that the sorption process does not cause any substantial structural responses of the framework, as demonstrated by in situ PXRD experiments under $CO_2$ atmosphere at 195 K (Supplementary Methods 10.1), we examined the correlation between the pore volumes determined from the $CO_2$ sorption isotherms and the volumes theoretically occupied by the alkoxy groups (here in terms of their Connolly solvent-excluded volume, Supplementary Methods 9.2). With the inclusion of MOF-5, a non-linear relationship becomes apparent, since the experimental pore volumes of the non-crystalline derivatives with shorter alkoxy substituents (C2–C6) are significantly lower than expected from linear extrapolation (Fig. 5b). Moreover, the deviation from the expected pore volume gets stronger with decreasing chain lengths. We ascribe this to increasing structural deformation (and thus contraction) when proceeding from dry-MOF-5-C6 to dry-MOF-5-C2. Based on the experimental pore volumes, we further estimate a volume contraction of the dry-MOF-5-CX materials compared to the cubic as-MOF-5-CX (Supplementary Table 9.3). The results are in good agreement to the values approximated from PXRD data (see above) and range from −4% for dry-MOF-5-C6 to −14% for dry-MOF-5-C2.

Application of $n$-butane as sorbate produces a strikingly different response of the materials, due to the stronger interaction of this gas with the hydrophobically lined interior of the pores (polarizability $\alpha$ ($n$-butane) = 8.02 Å$^3$ compared to $\alpha$($CO_2$) = 2.51 Å$^3$)[66] and the higher measurement temperature (293 K). Dry-MOF-5-C7 displays a stepped $n$-butane sorption isotherm with a slight hysteresis, which signifies a rhombohedral-to-cubic transition at a $n$-butane pressure of 8.0 kPa ($p/p_0 \approx 0.04$) (Fig. 5c). This is confirmed by in situ PXRD experiments under variable $n$-butane pressure at 298 K, although the transition is shifted to a much higher pressure (50–100 kPa, $p/p_0 \approx 0.2$–0.4), potentially due to too short equilibration times during the in situ PXRD study (Supplementary Fig. 10.9). The in situ PXRD data further clarify that the breathing transition of dry-MOF-5-C7 during $n$-butane sorption is reversible once the $n$-butane pressure is reduced.

In contrast, dry-MOF-5-C6 features a simple Langmuir-shaped $n$-butane sorption isotherm without any obvious steps at 293 K. Here, in situ PXRD at 298 K demonstrates that the material continuously transforms from the non-crystalline distorted phase to the crystalline cubic phase with increasing $n$-butane pressure. After pressure release, dry-MOF-5-C6 does not retransform to the non-crystalline phase on the timescale of the PXRD data collection (10 min), but largely retransforms within 17 h (Supplementary Fig. 10.9). Similar to the thermal transition of dry-MOF-5-C6, the retransformation appears to be a gradual relaxation process. The behaviour of dry-MOF-5-C5 is

comparable to that of dry-MOF-5-C6, but the structural changes are of a smaller extent (Supplementary Fig. 10.7).

Most surprisingly, propylene and propane sorption on dry-MOF-5-C6 and dry-MOF-5-C7 at 293 K indicate a clear breathing transition of both of these frameworks upon adsorption of these gases (Fig. 5d). For dry-MOF-5-C7 again small steps and hystereses ascribed to the rhombohedral-to-cubic transition are evident. Dry-MOF-5-C6 features much larger steps in the propane and propylene isotherms at ~15 kPa and strong hystereses, thus demonstrating that the non-crystalline-to-crystalline transition of this framework is triggered once a certain threshold pressure is reached.

Generally, these data support our hypothesis that guests, which exhibit favourable interactions with the DEDs decorating the frameworks are able to induce a structural response by counter-balancing the molecular forces leading to the non-correlated deformation of the framework architecture.

**Frustrated flexibility in other MOFs**. On the basis of our findings and analyses, we aimed for the extension of the phenomenon of frustrated flexibility towards other MOF structures. For that purpose, we prepared an isoreticularly expanded derivative of MOF-5-CX, namely IRMOF-10-C8, featuring an alkoxy-functionalized 4,4′-biphenyldicarboxylate linker, and a MOF of the entirely different **qom** topology[67], MOF-177-C8, featuring an analogously functionalized 1,3,5-benzene-tris(4′-benzoate) linker (Supplementary Methods 1.3 and 1.8). Crucially, these materials also convert from the crystalline state (as, guest-filled) into a non-crystalline, distorted state upon guest removal and return to the crystalline state when $N,N$-diethylformamide (DEF) is added (Supplementary Methods 1.9 and Figs. 2.14 and 2.15). We thus conclude that not only MOF-5-based materials, but also expanded versions of MOF-5, as well as a great number of other MOFs exhibiting the prominent [$Zn_4O(O_2C)_6$] node, may show stimuli-responsive frustrated flexibility as presented here. We notice that some of the recently reported polyMOFs likewise show a dynamic loss and recovery of crystalline order in response to guest molecules[68]. In principle, frustrated flexibility may be a frequent, unrecognized feature of a number of other MOF structure types, which so far are deemed to be structurally rigid but could straightforwardly be functionalized with suitable DEDs, such as UiO-66/67/68 or MIL-101 materials.[50,52]

Moreover, we suggest that frustrated flexibility, i.e. the incompatibility of intra-framework forces with the geometrical constraints of the MOF's building units, could be a fundamental principle behind the often reported amorphization of MOFs upon guest removal.[27,69–71] Utilization of organic building units, carrying precisely designed additional functional groups, is one of the major synthetic advantages of MOFs when compared to other porous materials. Our results demonstrate that such additional functionalities can have a major impact on the material's stimuli-responsive phase behaviour by changing the balance between enthalpic and entropic contributions to the free-energy landscape of the MOF. In particular, these contributions to the free-energy landscape are specific to a number of external conditions, which offers possibilities to gradually tune responsive properties to the desired application.

## Discussion

Herein, we reported a series of alkoxy-functionalized MOF-5 derivatives, which exhibit frustrated framework flexibility as a function of guest-loading and temperature. Structural distortion as a consequence of intra-framework dispersion interactions only occurs in a concerted and thus crystalline manner if the volumetric changes of the flexible frameworks are small (up to ~3%). Larger contractions (up to ~17%) require random distortions and the loss

of crystalline order. The frustration of the frameworks arises from the conflict of the framework lattice, which is rigid due to intrinsic geometrical constraints of the inorganic building units, with intra-framework dispersion interactions, demanding a densification of the structure. Unprecedentedly, the non-crystalline phases of these materials can be reversibly converted to the crystalline phase not only by adsorption of specific guest molecules (driving force: adsorption enthalpy), but also by heating to elevated temperature (driving force: entropy). The phenomenon of frustrated flexibility has consequences for the application of normally rigid MOFs in gas storage, separation and catalysis, and further suggests great potential for the discovery of new responsive materials exhibiting unconventional and exotic properties.

## Methods

**Materials synthesis**. Synthetic procedures for the organic linkers and the corresponding MOFs are given in the Supplementary Information.

**Powder X-ray diffraction (PXRD)**. Powder X-ray diffraction data of as-synthesized, dried and resolvated materials were collected on a Siemens D5005 diffractometer in Bragg-Brentano geometry using CuKα radiation in a range from 2.5° or 5.0° to 50° 2θ with a step size of 0.02°. All samples were finely ground and placed on either a glass holder or a zero-background sample holder made of single crystalline silicon (cut along the (610)-plane) or, if mentioned, on a holder made of plastic. Structureless profile fitting (Pawley method[72]) was performed using the routines provided by the TOPAS-*academic v6* software package[73]. Further diffraction data were collected at beamline P02.1 at DESY (Deutsches Elektronen-Synchrotron, Hamburg, Germany) with a monochromatic X-ray beam ($\lambda = 0.2073$ Å) using a Perkin Elmer XRD1621 flat panel detector. The samples were finely ground and measured using 0.7 or 1.0 mm diameter quartz capillaries. Data were integrated using the DAWN[74,75] software package.

**Single crystal X-ray diffraction (SCXRD)**. Single crystal X-ray diffraction data (SCXRD) for all as-synthesized compounds and MOF-5-C3/C6 resolvated with *n*-octane were collected on either an Oxford Diffraction Xcalibur or a Bruker D8 Venture diffractometer using either MoKα ($\lambda = 0.7107$ Å) or CuKα ($\lambda = 1.5418$ Å) radiation, further equipped with an Oxford liquid nitrogen cryostream to keep the sample environment at 100 or 250 K. The raw data was processed with the APEX3 suite or the CrysAlisPro software package. The structure solution and refinement processes were conducted with the Olex2[76] interface using SHELXS, SHELXT and SHELXL[77].

**Solution NMR spectroscopy**. $^1$H and $^{13}$C NMR spectroscopy was performed on digested MOF samples (in DMSO-$d_6$ with DCl/$D_2$O, 35% weight, <0.1 ml)) and synthesized organic linkers (DMSO-$d_6$ or CDCl$_3$) utilising Bruker DPX-300, DPX-500 or Agilent DD2 500 spectrometers. The data were processed using the ACD/Labs software. $^1$H and $^{13}$C NMR spectra were referenced to the respective residual solvent signal and chemical shifts are given with respect to tetramethylsilane.

**X-ray pair distribution function (XPDF) analysis**. X-ray total scattering experiments were performed at beamline I15-1 at DLS (Diamond Light Source, UK) using a monochromatic X-ray beam with an energy of 76.7 keV ($\lambda = 0.161669$ Å). Finely ground samples were loaded into 1.5 mm (outer diameter) borosilicate glass capillaries. Scattering data from an empty capillary was used for background subtraction. Corrections for background, multiple, container and Compton scattering as well as absorption were done with the GudrunX program. The pair distribution functions on the form of $D(r)$ were obtained via Fourier transform of the normalized reciprocal space data $S(Q)$ [78,79].

For the as-synthesized materials, the correct chemical compositions were determined by use $^1$H NMR spectroscopy, since the pores of the MOF-5-CX are filled with a number of DMF molecules (Supplementary Methods 1.4).

**IR spectroscopy**. IR spectra were recorded on a Perkin Elmer Spectrum Two Fourier transform IR spectrometer ($\tilde{\nu} = 400$–4000 cm$^{-1}$) in reflection mode using a diamond ATR (attenuated total reflectance) unit. All measurements were conducted under ambient conditions. As-synthesized samples were shortly dried on a filter paper to remove surface DMF. For *n*-octane reinfiltrated samples, a small drop of *n*-octane was added after loading the sample onto the diamond crystal to avoid solvent evaporation from the pores of the compounds under investigation.

**Solid state NMR spectroscopy**. Solid state NMR experiments were carried out using a 2.5 mm triple-resonance ($^1$H, $^{13}$C, $^{15}$N channels) MAS probe on a 700 MHz Bruker Avance Neo spectrometer and a spinning frequency of 33.33 kHz at an overall effective sample temperature of 5 °C. The temperature calibration was performed externally with KBr powder.[80] For 1D $^{13}$C cross-polarisation NMR

experiments, data acquisition was performed for 19.93 ms with a recycle delay of 3 s and total number of scans of around 6000–7000. The applied decoupling sequence was swept-frequency two-pulse phase modulation.

**Thermal analysis**. Simultaneous thermogravimetric analysis and differential scanning calorimetry (TG-DSC) experiments were performed using a STA504 by TA Instruments under a constant Argon flow (4 l h$^{-1}$). Measurements for all as and dry MOFs were carried out from 30 to 770 °C with a heating rate of 10 K min$^{-1}$. Further DSC experiments were performed using a TA Instruments DSC-25 under a constant N$_2$ flow (50 ml min$^{-1}$). ΔH of MOF-5-C7 and MOF-5-C8 were extracted from experimental data (up- and downscans) collected over a temperature range from 25 to 250 °C with a heating/cooling rate of 10 K min$^{-1}$. ΔH was determined as the integral area of the corresponding calorimetric peak following baseline subtraction.

**Variable temperature PXRD (VT-PXRD)**. Powder X-ray diffraction data at various temperatures were collected at beamline P02.1 at DESY (Deutsches Elektronen-Synchrotron, Hamburg, Germany) with a monochromatic X-ray beam ($\lambda = 0.2073$ Å) using a Perkin Elmer XRD1621 flat panel detector. All samples were finely ground, filled into 0.7 or 1.0 mm diameter quartz capillaries and heated or cooled using a combined setup of an Oxford CryoStream ($T \leq 500$ K) and an Oxford HotAirBlower (from $T > 500$ K). Temperature calibration was conducted by reference PXRD measurements using Al. Data were integrated using the DAWN software package. Further experiments of the samples dry-MOF-5-C2 and dry-MOF-5-C6 were performed at BL9 of DELTA (Dortmunder Elektronenspeicherring-Anlage, Dortmund, Germany) with a monochromatic X-ray beam ($\lambda = 0.6199$ Å) using a MAR345 image plate detector. Finely ground samples were sealed in quartz capillaries and heated using an Oxford CryoStream in the range from 300 to 500 K. Data were integrated using the DAWN software package[74,75].

**Isothermal gas sorption**. Sorption experiments were undertaken with a Quantachrome Autosorb iQ MP porosimeter using only high-purity adsorptive gases (N$_2$: 99.999%, CO$_2$: 99.995%, *n*-butane: 99.95%, propane: 99.95%, propylene: 99.95%). Sample quantities of at least 40 mg were used for the experiments. Prior to the measurements each sample was carefully grinded and afterwards degassed in dynamic vacuum ($p \approx 10^{-5}$ kPa) at 100 °C. Sorption isotherms were measured for the gases N$_2$ and CO$_2$ at 77 K and 195 K and for *n*-butane, propane and propylene at 293 K, respectively.

**In situ CO$_2$ sorption PXRD**. Collection of powder X-ray diffraction data as a function of CO$_2$ pressure was conducted at beamline P02.1 at DESY (Deutsches Elektronen-Synchrotron, Hamburg, Germany) with a monochromatic X-ray beam ($\lambda = 0.2073$ Å) using a Perkin Elmer XRD1621 flat panel detector. The samples were finely ground, filled into 0.7 or 1.0 mm diameter quartz capillaries and mounted onto a custom-made vacuum-tight gas cell. The gas cell setup did not allow to spin the capillaries so that the data were collected statically without sample spinning. During the experiments all samples were kept at 195 K using an Oxford liquid nitrogen CryoStream. Measurements were performed within a pressure range of 10$^{-2}$ (vacuum) and 100 kPa CO$_2$. High-purity CO$_2$ (99.995%) was used. At each pressure step the samples were equilibrated for 10–15 min.

**In situ *n*-butane sorption PXRD**. Collection of powder X-ray diffraction data as a function of *n*-butane pressure was conducted at beamline BL9 at DELTA (Dortmunder Elektronenspeicherring-Anlage, Dortmund, Germany) with a monochromatic X-ray beam ($\lambda = 0.4600$ Å) using a MAR345 image plate detector. The samples were finely ground, filled into 1.0 mm diameter quartz capillaries (but not sealed) and placed in a custom-made vacuum-tight gas cell. The gas cell setup did not allow to spin the capillaries so that the data were collected statically without sample spinning. All measurements were performed at room temperature (about 293 K) using *n*-butane of a purity of 99.5%. At each pressure step the samples were equilibrated for 2–10 min depending on the sample. For dry-MOF-5-C2 to dry-MOF-5-C6 equilibration times of 10 min per pressure step were applied, whereas dry-MOF-5-C7 and dry-MOF-5-C8 were equilibrated for only 2 min at each step (step size 50 kPa).

**Microscopy**. Microscopy images were taken with a Leica DM2500LED polarisation microscope including camera and Linkam LTSE420 temperature stage using the Basler Microscopy Software.

**Computational details**. All simulations were performed with the LAMMPS[81] program package using our in-house developed *pylmps* python wrapper to drive the simulations. An integration step size of 1 fs was used for all simulations. If not noted otherwise, thermostat relaxation times of 100 fs and barostat relaxation times of 1 ps were utilized. The direct Ewald summation was used for long-range coulomb interactions using a relative force accuracy of $1 \cdot 10^{-6}$. The large simulations were performed using the particle-particle particle-mesh long-range coulomb solver for speedup using the same accuracy. The barostat implementation for the $NP(\sigma_a = 0)T$

and $NV(\sigma_a = 0)T$ ensemble simulations are taken from GitHub (https://github.com/stevenvdb/lammps/tree/newbarostat). PXRD patterns were calculated using Fox (https://fox.vincefn.net/) using a small Lorentzian peak shape width of 0.001° and CuKα XPDFs were calculated using the diffpy (www.diffpy.org) python package utilising an instrumental Q resolution factor of qdamp = 0.05. The $p(V)$ equation of state was fitted using a polynomial of degree $n$, where $n$ was chosen manually to assure no apparent under- or overfitting (see plots in Supplementary Fig. 11.4 for details). $\Delta A$ was calculated by numerical integration of $p(V)$ and $\Delta U$ by averaging the total potential energy along the respective trajectory.

## Data availability

The experimental data supporting this study are included in this published article and the associated supplementary information document. The raw data are available from the corresponding author upon request. The X-ray crystallographic coordinates for structures reported in this study have been deposited at the Cambridge Crystallographic Data Centre (CCDC), under deposition numbers 2040916-2040925. These data can be obtained free of charge from The Cambridge Crystallographic Data Centre via www.ccdc.cam.ac.uk/data_request/cif. Initial structures, force field parameters and input files necessary to rerun the MD calculations are available at https://github.com/cmc-rub/supporting_data/tree/master/90-Pallach-Keupp-NatCommun.

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

## Acknowledgements

Funding from TU Dortmund (R.P. and S.H.), DFG SPP 1928 COORNETs Start-up Grant (S.H.) and the *Fonds der Chemischen Industrie* Grant Number 661602 (S.H.) is gratefully acknowledged. R.S. and J.Ke. would like to thank for financial support within the research unit FOR 2433 "Switchable MOFs" (Grants SCHM 1389/10-1 and 2). Funded by the Deutsche Forschungsgemeinschaft (DFG, German Research Foundation) under Germany's Excellence Strategy—EXC 2033 – 390677874—RESOLV. We thank DESY (Hamburg, Germany), a member of the Helmholtz Association HGF, for the provision of experimental facilities. Parts of this research were carried out at PETRA III on beamline P02.1 (experiments I-20180709 and I-20180710). This work was carried out with the support of Diamond Light Source, beamline I15-1 (proposals EE15895-1, EE21262-1 and CY21604-2). The research leading to this result has been supported by the project CALIPSOplus under the Grant Agreement 730872 from the EU Framework Programme for Research and Innovation HORIZON 2020. We further acknowledge the DELTA machine and BL9 staff for support during our experiments.

## Author contributions

S.H. designed and led the project. R.P. synthesized the organic linkers and the MOFs and collected and analysed PXRD, SCXRD, IR spectroscopy, solution NMR spectroscopy, TGA/DSC, optical microscopy and gas sorption data. K.T. contributed to the organic linker syntheses. R.P., L.F.-B., P.A.C. and S.H. performed the X-ray total scattering experiments. R.P., L.F.-B., M.K., M.T.W. and S.H. performed the variable temperature PXRD experiments. R.P., L.F.-B., A.M., M.T.W. and S.H. performed the in situ PXRD experiments under CO2 sorption. R.P., L.F.-B., M.K., C.S. and S.H. performed the in situ PXRD experiments under n-butane sorption. J.Ke. performed the MD simulations under supervision of R.S. J.Ko. and S.K.V. performed and analysed the solid-state NMR experiments under supervision of R.L. All authors participated in discussing the data. R.P. and S.H. wrote the paper with J.Ke. and R.S. contributing the section on molecular modelling.

## Funding

## Competing interests

The authors declare no competing interests.

## Additional information

**Peer review information** *Nature Communications* thanks Simon Krause and other, anonymous reviewers for their contirbutions to the peer review of this work. Peer review reports are available.

