## [Peer Review File · Nature Communications]

REVIEWER COMMENTS

Reviewer #1 (Remarks to the Author):

In this manuscript, Pallach and co-workers demonstrate the targeted design of frustrated flexibility in metal-organic frameworks (MOFs) by systematically introducing longer alkoxy chains as dispersion energy donors (DEDs) in the otherwise rigid MOF-5 (proof of concept material), IRMOF-10, and MOF-177. These new materials show a variety of exciting flexible behavior between crystalline and amorphous phases, as characterized extensively by (single-crystal) X-ray diffraction, nuclear magnetic resonance, pair distribution functions, infrared spectroscopy, and gas sorption experiments. The MOF-5 analogues with the shortest alkoxy chains are demonstrated to undergo a crystalline-to-amorphous transition from their cubic phase upon drying. In this transition, the inorganic building block is distorted in a random fashion throughout the framework to maximize the dispersion interactions between the alkoxy chains although the framework connectivity and topology is retained. Surprisingly, the amorphous materials recover their crystalline order after resolution with octane or DMF. For the MOF-5-C7 material, the longer alkoxy chain yields a correlated distortion of the cubic framework upon drying, resulting in a crystalline rhombohedral crystal that can be retransformed into the original cubic crystal upon resolution. The molecular dynamics simulations clearly show the influence of the DEDs on the formation of different (meta)stable states that become accessible, although in contrast to the experiments these low volume states only become accessible when a pressure is applied to the material. Finally, the authors also show that the recovery of the crystalline order can also occur by heating the amorphous materials and characterize the sorption properties of these MOFs.

In my opinion, this study is a very valuable and clearly written contribution that helps understanding the formation of (non-)crystalline phases in MOFs and how to design them via frustration. The work combines a large variety of well-executed experimental and computational tools that confirm the loss and subsequent recovery of crystallinity in MOFs with frustrated flexibility and it highlights the design possibilities of this approach by considering different MOFs. Based on this assessment, I would suggest publication of a revised form of this manuscript in Nature Communications, taking into account the remarks given below.

1. Do the authors have an idea of the energy needed to deform the inorganic node in the MOF-5 material (necessary to access either crystalline rhombohedral as well as the amorphous phases)? Can they compare this energy to the energy associated with the interactions induced by the DEDs?
2. On page 19, it is mentioned that materials typically lose crystallinity with rising temperature. I'm unsure whether this statement is also generally true for materials that undergo second-order phase transitions. For instance, in perovskites, higher temperatures may promote the existence of more ordered phases and hence a larger crystallinity as higher temperatures remove the inequivalency of different configurations by enabling transitions between them. It might be worthwhile to investigate whether this could also be the case for the materials discussed here.
3. The authors observe a distinct change when considering the as-MOF-5-CX materials with $X \leq 7$ (cubic) versus the one with $X = 8$ (rhombohedral). Did they also attempt to synthesize MOF-5 analogs with even longer alkoxy chains to see whether the same behavior as for $X = 8$ is obtained?
4. The volumes of the metastable states of MOF-5-CX as predicted by simulation are substantially smaller than the experimental dry-MOF-5-CX samples, as acknowledged by the authors. This is attributed to the smaller box size (2x2x2) and insufficient sampling of the linker configurations. Did the authors observe a better reproduction of the volumes for the 8x8x8 models they considered for MOF-5-C3 and MOF-5-C6?
5. Although the text mentions the absence of long-range correlation between the volumes of adjacent cells after transformation to the metastable phase in Figure 4d-e, it seems there is some stronger

correlation along certain directions in Figure 4e, namely along the out-of-plane and the vertical in-plane directions. This seems very similar to earlier observations by some of the authors in the breathing phase transformation in DMOF-1 (10.1002/adts.201900117), as well as by others (10.1038/s41467-019-12754-w). Can the authors comment on whether this correlation observed here is also found in their other MOF-5-CX simulations and whether it is similar to the correlation in those other flexible materials?

6. On page 15, the authors mention that MOF-5-C3 to MOF-5-C6 feature a metastable contracted state. From Figure 4a, however, this seems to be the case also for MOF-5-C2. Is this correct and, if so, is there a reason not to mention MOF-5-C2 along with MOF-5-C3 to MOF-5-C6?

7. I fully agree with the authors that frustrated flexibility may be present but overlooked in other MOF structures. However, given that the frustrated flexibility in the materials studied here is intimately connected with the tetrahedrally symmetric Zn₄O building block that occupies a position of octahedral symmetry, I'm unsure how this concept translates to the other MOFs mentioned here, such as the UiO-66 series, MIL-100/-101, and the HKUST-1 material. To enable the design possibilities, it may be worthwhile to discuss this in further detail.

8. In Figure S10.1, it seems that the nitrogen isotherm drops at higher pressures for dry-MOF-5-C2. Is this correct? If so, what is the origin of this drop?

9. For the temperature ramp simulations discussed in section S12, it seems the total simulation time and the ramp conditions are missing.

10. Are the differences in p(V) curves when using initial states obtained either from the pressure ramp or the temperature ramp simulations connected with the range of p(V) curves that can be obtained when having large sidechain functionalization (see for instance 10.1002/anie.202011004), or is this a completely separate effect?

11. Can the authors pinpoint the origin of the large deviations with respect to the fitting curve for MOF-5 in Figure S12.4?

12. In the caption of Figure S12.6, the authors explain the significance of yellow, magenta, and dark blue boxes. However, it seems no explanation is given of the meaning of the light blue boxes.

Reviewer #2 (Remarks to the Author):

In „Frustrated flexibility in metal-organic frameworks“ Pallach et al. demonstrate the deformation of “rigid” MOF-5 derivatives that contain alkoxy side-chains, referred to as dispersion energy donors (DEDs) attached to the ligand backbone. The work is based on extensive experimental and computational studies and heavily builds on previous work of the authors about entropy-controlled deformation in flexible MOFs. It addresses the interesting situation in which internal dispersion interactions overcome the energy barrier for structural deformation of the framework lattice. In this work the authors utilize the MOF-5 framework with pcu topology which is in particular interesting as it is widely perceived as a rigid and non-deformable framework (topology).

I consider the paper a very interesting piece of work which I enjoyed reading. It certainly represents an important contribution to the field of flexible MOFs / soft porous crystals. The paper is well written and illustrated, uses appropriate citations and I highly recommend publication in Nature Communications considering the following suggestions which are of curious nature:

P3117 – with respect to crystalline – amorphous-crystalline transitions I would like to highlight recent work on mesoporous flexible MOFs (<https://pubs.rsc.org/en/content/articlelanding/2020/sc/d0sc03727c>). It seems that with extended

framework backbone structural contraction occurs under enhanced loss of long-range order. The authors are free to cite the work or not. Current citations already seem appropriate.

P411 – What about strong covalent bonds of the ligand itself? Hinging and buckling deformations of the ligand itself is an additional degree of freedom that can contribute to the deformation of the whole lattice? (this might actually become more relevant when the ligands are elongated).

P414 “distorted MOF phases exhibit the same topology” – I am no expert in the field and definition of topologies but can a distorted lattice be named topology? I would just state “same connectivity and bonding pattern”.

P4122 “At first, we select an intrinsically rigid and non-responsive MOF structure type, which does not allow for correlated (and thus crystalline-to-crystalline) large-magnitude structural changes.” – Now this to me is a difficult statement because it lacks a clear definition of a large magnitude change. Staying with MOF-5 there are now variants of this framework (<https://pubs.acs.org/doi/abs/10.1021/jacs.5b11150>) that are structurally/symmetrically different than the original cubic phase. So maybe the authors can either differentiate their definition of large-magnitude against previous examples of deformations in the MOF-5 system, or at best provide a quantitative figure (let’s say for example 5% reduction in unit cell volume). ◊ I now see that the authors extensively quote on that later in the manuscript, still a quantification (like you provide on page 718) of “large” would be nice.

P4126 “thus change the energy landscape of the MOF structure.” – here I would clearly define which parameters define the energy landscape. In this case looking at the performed computation energy vs. unit cell volume and hydrostatic pressure.

P4128 with respect to frustration is there any kind of kinetic idea/concept associated to this state?

P5124 “since they would require major distortions of its inorganic nodes.” I would add bond breaking here which in a coordination cluster might also occur reversibly (e.g. transmetalated MOF-5 derivatives by Dinca et al.) – I would also cite Ferey et al. here (<https://pubs.rsc.org/en/content/articlelanding/2009/cs/b804302g#!divAbstract>) as they were really the first to make that comparison

P10124 “Remarkably, the mosaicity of the crystals does not increase after the guest-mediated crystalline-to-amorphous-to-crystalline transition. This indicates that the loss of crystallinity in dry-MOF-5-CX occurs mainly through a strain-based process, rather than cracking or bond breaking.” ◊ Very elegant!

With respect to the recovery of crystallinity do the authors have any idea to what extent the guest species contribute to the scattering? Unfortunately, there is no information provided in the ESI or manuscript if e.g. a squeeze method was performed after SCXRD refinement.

P1316 “2 x 2 x 2 cavities in dry-MOF-5-C6.” – does that correspond to 2 x 2 x 2 unit cells? Maybe wise to use crystallographic description for a better correlation with the crystallographic results?

P14118 “In the smaller 2x2x2 simulation cells” – I assume this now correlates to unit cells?

P14124 I think at this point it is important to mention that the simulations were carried out on guest-free crystal structures. Maybe apply your dry-terminology as previously in the text. I think this is an important fact because in the real-world system the dry system is obtained by solvent removal and not directly made in the solvent-free state. In particular with respect to the presence and population of metastable states on the energy surface this is crucial.

Fig 4. Out of interest: Has the energy landscape of MOF-5 without DED been calculated? Could be a

good reference to probe the functionality of the force field.

Also maybe try to reference the 0 energy for a and b the same way being the energy minimum for the expanded large volume state. This way it might be easier to see the degree of stabilization for free vs. internal energy. Just a suggestion for better visualization.

The ability to match X-ray scattering with the simulated distorted structures is fantastic! Congrats. P17|12 "The dispersion interactions introduced by the DEDs are apparently strong enough to outbalance the energetic cost for the deformation of the nodes" – I agree. Is it possible to derive the energetic costs from the MD simulations? For example which bond length elongation or dihedral deformation are the energetically most demanding distortions? – Can be material for another paper and I don't consider it crucial for this particular study but certainly of interest in particular since the calculations have been performed.

P18|19 "The thermal behaviour of dry-MOF-5-C6 is counterintuitive and inverse to the thermal behaviour of conventional materials." – I am wondering if there are any comparable cases to this kind of behavior? I consider it quite outstanding, in particular given the stark contrast between MOF-5-C6 and MOF-5-C7

P20|4 "whereas all other derivatives only show N2 4 adsorption on the external surface" I recommend using values of MOF-5 as a reference (can be published values) to illustrate the degree of pore occupation by DED. – As you do later for CO2 at 195 K

Figure S10.1 – black isotherm exhibits a negative slope which I assume is caused by inaccurate dead volume measurement. I suggest repeating the experiment or at least commenting on this! I can understand that the quantities required to accurately measure nitrogen isotherms for materials with such low porosity are difficult to achieve on this class of solids. A statement in the ESI will put the isotherms in perspective and avoid conflicts with the adsorption community.

Fig. 5b Why would the authors expect a linear correlation? This is not clear to me neither from the figure nor the text. Also I could not find detail. Also be advised that a 1.5% change in unit cell volume can likely

I find the correlation/analysis of pore volume and cell contraction somewhat suitable but I am not sure if for example inaccessibility of pore space due to the DED could manipulate the adsorption experiments to a degree not detectable. The observed correlation might indicate that this is not the case, however I am unsure which "undetected" contributions could also be present.

I suggest giving the respective temperatures (p20|23) to the adsorption probes, not just in the first sentence but also further down in the text in particular for butane where temperature between adsorption and PXRD experiment varies. We have recently demonstrated the effect of temperature in DUT-49 (<https://pubs.rsc.org/en/content/articlelanding/2021/fd/d0fd00013b>) and believe these factors will be valid for most flexible or even frustrated MOFs. The statement "due to the stronger interaction of this gas with the hydrophobically lined interior of the pores 24 (polarizability α (n-butane) = 8.02 Å³ compared to α (CO₂) = 2.51 Å³)" is thus only partially accurate as the solid-fluid interactions are also a function of temperature relative for each respective guest.

P21|1 Is it only a shift towards higher absolute pressure or also a shift to higher relative pressures. The latter would indicate a change in thermodynamics of the solid-fluid interactions, while a shift in absolute pressure is to be expected due to the change in temperature.

P21|2 "due to too short equilibration times" – is there any experimental support for this claim and what are the equilibration times?

P21|4 "the non-crystalline distorted phase to the crystalline cubic phase with increasing n-7 butane pressure." Previously in the paper the authors demonstrate reversible transitions by suspension in n-octane. I suggest to correlate the observations of this suspension and provide timescale required for these transformations (e.g 2 days in octane vs. 10 min equilibration for n-butane adsorption)

P21|8 "After pressure release, MOF-5-C6 does not immediately show retransformation to the non-crystalline phase." Can you provide a time that you consider "not immediate"? Quantification would be very helpful as kinetic processes in framework deformation are currently intensely discussed.

Can the authors provide cif files of the MOF materials and the models used for computation as

supplementary information?

I want to express my support for submitting such an extensive study in a single paper! I think this work could have easily been split up in 2-3 individual papers but a lot would have been lost in that process! The paper really benefits from the large body of experimental and computational work! Congrats and well done managing such an extensive volume of work!

In 2015 at a Symposium on flexible MOFs in Paris (to which some of the authors attended as well) Gérard Férey insisted after a lively discussion on flexibility that MOF-5 cannot be flexible. The proposed concept of "frustrated flexibility" described in this work illustrates that we have a long way to go in knowing what can and what cannot be "flexible". Well done!

Lastly, a comment:

While I do in general agree to the term of frustrated flexibility in the case described here, I would be careful applying it/extending it to all kinds of responsive flexibility in framework materials. I would associate frustration in the given context to some kind of kinetically trapped state which to some extent is present in the investigated system. In the future I would hope for a more concise and extended definition of such framework frustration. In essence, any framework material can be compressed in a denser state by application of hydrostatic pressure. Whether this can occur reversibly mainly depends on the absence of framework disintegration. The uniqueness of the present system (in my opinion) arises from the temperature responsiveness. It also seems that the presence of DED overcomes a lot of issues associated with framework disintegration and furthermore overcomes issues associated with guest induced transitions (e.g. diffusion, concentration gradients, accessibilities, etc.). They seem to be a perfect system to systematically study entropic effects.

Dr Simon Krause

RESPONSE TO REVIEWER COMMENTS

Reviewer #1 (Remarks to the Author):

In this manuscript, Pallach and co-workers demonstrate the targeted design of frustrated flexibility in metal-organic frameworks (MOFs) by systematically introducing longer alkoxy chains as dispersion energy donors (DEDs) in the otherwise rigid MOF-5 (proof of concept material), IRMOF-10, and MOF-177. These new materials show a variety of exciting flexible behavior between crystalline and amorphous phases, as characterized extensively by (single-crystal) X-ray diffraction, nuclear magnetic resonance, pair distribution functions, infrared spectroscopy, and gas sorption experiments. The MOF-5 analogues with the shortest alkoxy chains are demonstrated to undergo a crystalline-to-amorphous transition from their cubic phase upon drying. In this transition, the inorganic building block is distorted in a random fashion throughout the framework to maximize the dispersion interactions between the alkoxy chains although the framework connectivity and topology is retained. Surprisingly, the amorphous materials recover their crystalline order after resolution with octane or DMF. For the MOF-5-C7 material, the longer alkoxy chain yields a correlated distortion of the cubic framework upon drying, resulting in a crystalline rhombohedral crystal that can be retransformed into the original cubic crystal upon resolution. The molecular dynamics simulations clearly show the influence of the DEDs on the formation of different (meta)stable states that become accessible, although in contrast to the experiments these low volume states only become accessible when a pressure is applied to the material. Finally, the authors also show that the recovery of the crystalline order can also occur by heating the amorphous materials and characterize the sorption properties of these MOFs.

In my opinion, this study is a very valuable and clearly written contribution that helps understanding the formation of (non-)crystalline phases in MOFs and how to design them via frustration. The work combines a large variety of well-executed experimental and computational tools that confirm the loss and subsequent recovery of crystallinity in MOFs with frustrated flexibility and it highlights the design possibilities of this approach by considering different MOFs. Based on this assessment, I would suggest publication of a revised form of this manuscript in Nature Communications, taking into account the remarks given below.

1. Do the authors have an idea of the energy needed to deform the inorganic node in the MOF-5 material (necessary to access either crystalline rhombohedral as well as the amorphous phases)? Can they compare this energy to the energy associated with the interactions induced by the DEDs?

Comment: We do not have data on strained aperiodic cluster models readily available, but what we do have is a scan of the total energy of the parent MOF-5 material (calculated with MOF-FF) as a function of cell angles ranging between 90 ° and 100 °. Due to the delicate balance between enthalpy and entropy it is not possible to pinpoint the DED induced interactions. However, from the parent MOF-5 deformation energy and the computed enthalpy profiles (Fig. 4b) together with the experimentally observed deformation angle, a necessary energy gain of at least about 12 kJ/mol for the case of MOF-5-C8 can be estimated per formula unit for the attractive DED interactions. We have added a short paragraph and figure (Supplementary Figure 11.14) to the SI where we discuss this topic.

2. On page 19, it is mentioned that materials typically lose crystallinity with rising temperature. I'm unsure whether this statement is also generally true for materials that undergo second-order phase

transitions. For instance, in perovskites, higher temperatures may promote the existence of more ordered phases and hence a larger crystallinity as higher temperatures remove the inequivalency of different configurations by enabling transitions between them. It might be worthwhile to investigate whether this could also be the case for the materials discussed here.

Response: It is generally true that the absolute intensity of all Bragg reflections of any crystalline material is attenuated with increasing temperature due to increased thermal motion. This fundamental observation has been explained by Peter Debye (DOI: 10.1002/andp.19133480105) and was further refined by Ivar Waller (DOI: 10.1007/BF01328696). In crystallography this phenomenon is accounted for by the so-called Debye-Waller factor, which describes the attenuation of diffracted X-rays as a result of thermal motion. The observation of a decreased intensity of diffracted X-rays is also generally true for materials showing 2nd order phase transitions. In the classical BaTiO₃ perovskite, the series of phase transitions from rhombohedral to orthorhombic to tetragonal to cubic with increasing temperature is a consequence of order-disorder transitions involving the displacement of the Ti^{IV} cations. This means, the cubic high-temperature phase of BaTiO₃ certainly exhibits a higher symmetry than the preceding tetragonal phase, but at the same time a lower order (i.e., crystallinity) due to the entropically favoured maximized disorder of Ti^{IV} cations over eight sites (see for example DOI 10.1080/00150199808009173). Thus, the thermal behaviour of BaTiO₃ is fundamentally different from the behaviour of *dry*-MOF-5-C6.

In our work, dispersive interactions between the pendent alkoxy groups and the framework backbone cause a structural contraction which requires distortions of the inorganic Zn₄O(CO₂)₆ building units. Due to the geometry and arrangement of these building units, the enthalpically stabilized contracted phases are X-ray amorphous (i.e., statically disordered). With increasing temperature, the dispersion interactions are counterbalanced by the increasing vibrational entropy of the alkoxy groups. Hence, a re-ordering of the framework backbone towards the crystalline cubic phase of *dry*-MOF-5-C6 can be observed in the form of increased Bragg scattering intensity and the shift of the first scattering peak towards lower scattering angles. In the crystalline higher temperature phase, the static disorder (configurational entropy) of the backbone is reduced, while the dynamic disorder (vibrational entropy) of the side chain increases. This is a unique case, where an increase in the total entropy of the system also results in an increased long-range order (crystallinity).

3. The authors observe a distinct change when considering the *as*-MOF-5-CX materials with X ≤ 7 (cubic) versus the one with X = 8 (rhombohedral). Did they also attempt to synthesize MOF-5 analogs with even longer alkoxy chains to see whether the same behavior as for X = 8 is obtained?

Response: That is an excellent question, which was also of our interest. Over the course of our investigations, we have indeed synthesized two more MOF-5-CX derivatives, namely MOF-5-C9 and MOF-5-C10. For the sake of conciseness and simplicity, these materials have not been included in the original version of the manuscript but have now been added as a new section to the Supplementary Information with all the data available at the moment (see Supplementary Methods 12). Interestingly, *dry*-MOF-5-C9 is rhombohedrally distorted similar to *dry*-MOF-5-C8, but to a lesser extent. Variable temperature PXRD experiments and DSC data clearly indicate that *dry*-MOF-5-C9 also transforms from the rhombohedral to the cubic phase when heated to ~160 °C. Unfortunately, the PXRD pattern of the solvated *as*-MOF-5-C9 is of poor quality, so that the rhombohedral peak splittings are not as clear as for *as*-MOF-5-C8. This might hint towards the simultaneous presence of the cubic and the rhombohedral phases in *as*-MOF-5-C9. Contrary, *as*- and *dry*-MOF-5-C10 both

possess a cubic structure. Thus, the bulk of the C10 alkyl chains appears to be too large, so that the pores are fully occupied in the cubic phase impeding a rhombohedral contraction upon guest removal.

4. The volumes of the metastable states of MOF-5-CX as predicted by simulation are substantially smaller than the experimental dry-MOF-5-CX samples, as acknowledged by the authors. This is attributed to the smaller box size (2x2x2) and insufficient sampling of the linker configurations. Did the authors observe a better reproduction of the volumes for the 8x8x8 models they considered for MOF-5-C3 and MOF-5-C6?

Response: The data the referee asks about is given in Figure 4f, where the average cell volume as well as its statistics (given by the 1 σ , 2 σ and 3 σ intervals) is plotted as a function of external pressure exerted to the system. The cell volume is the volume of a single $Zn_4O(CX-bdc)_3$ formula unit. The short answer to the question above is: No, also the large supercells exert an average cell volume, which is way too small compared to the average cell volumes estimated from the experimental PXRD patterns. However, we have to keep in mind here, that this particular simulation is at an elevated pressure. Extrapolating the low volume region back to ambient would result in a larger, yet too small average cell volume compared to the experiment. At this point it should be noted that also the experimental values are only estimates and that the model used to derive the average experimental cell volumes is targeting for the upper bound (i.e. the maximum cell volume and the minimum contraction). This is because the peak maxima of the asymmetric first scattering peaks have been used for the calculation and not the mean value. The mean values are shifted to higher scattering angles with respect to the peak maxima and thus would yield a smaller average experimental cell volume if applied in our simple calculation. Lastly, also the utilized MOF-FF is likely to contribute to the deviation: From test calculations for a molecular model of the linker side-chains (dimethoxyethane bulk liquid NPT simulations at standard conditions), we can conclude that the utilized MM3 vdW parameters are either a little too small or too attractive, resulting in an overall slightly overestimated density.

We believe, however, that the qualitative picture is what is important here, which is the large spread in individual cell volumes as well as the smaller degree of correlation compared to other MOF systems, for which such simulations have been carried out.

5. Although the text mentions the absence of long-range correlation between the volumes of adjacent cells after transformation to the metastable phase in Figure 4d-e, it seems there is some stronger correlation along certain directions in Figure 4e, namely along the out-of-plane and the vertical in-plane directions. This seems very similar to earlier observations by some of the authors in the breathing phase transformation in DMOF-1 (10.1002/adts.201900117), as well as by others (10.1038/s41467-019-12754-w). Can the authors comment on whether this correlation observed here is also found in their other MOF-5-CX simulations and whether it is similar to the correlation in those other flexible materials?

Response: The reviewer is absolutely right about this and we had omitted this point in the original manuscript to save space. There is indeed a non-negligible, yet much weaker correlation between neighboring cell volumes compared to the two papers mentioned above, as to where the phase transition starts and in which spatial regions correlated distortion takes place. Considering the cell volumes, layered domains are apparent only for MOF-5-C3 and not MOF-5-C6 (see Supplementary Figs. 11.9 and 11.10). In contrast to that, considering the cell angles, layered domains are clearly

visible for MOF-5-C3 and to some degree also in MOF-5-C6. This is even more apparent for the small cell simulations as can be seen in Supplementary Fig. 11.6. In this regard we may say that the transformation is similar to those mentioned in the references given by the reviewer. The difference here is the higher symmetry of the underlying topology, which allows for these layers to be present in all three spatial dimensions: In DMOF-1, there is only two distinct wine-rack modes, whereas herein, the three dimension each have two directions to where the distortion can take place. In addition, the paddle-wheel in DMOF-1 is much more flexible than the Zn_4O node, making this correlation much more feasible. Thus, a similar but weaker correlation in the cell angles can be observed, resulting in an overall weak correlation regarding the cell volumes. We have added a sentence in the main text to elaborate on this observation and added the respective references.

6. On page 15, the authors mention that MOF-5-C3 to MOF-5-C6 feature a metastable contracted state. From Figure 4a, however, this seems to be the case also for MOF-5-C2. Is this correct and, if so, is there a reason not to mention MOF-5-C2 along with MOF-5-C3 to MOF-5-C6?

Response: The reviewer is absolutely right about this. As it is obvious from Fig. 4a the numbers have erroneously been given shifted by one. We have corrected the numbers now from 'C3 to C6' to 'C2 to C5'.

7. I fully agree with the authors that frustrated flexibility may be present but overlooked in other MOF structures. However, given that the frustrated flexibility in the materials studied here is intimately connected with the tetrahedrally symmetric Zn_4O building block that occupies a position of octahedral symmetry, I'm unsure how this concept translates to the other MOFs mentioned here, such as the UiO-66 series, MIL-100/-101, and the HKUST-1 material. To enable the design possibilities, it may be worthwhile to discuss this in further detail.

Response: Indeed, in the current work we have only shown frustrated flexibility in a number of MOFs based on the tetrahedral Zn_4O building block. These MOFs have been selected as prototypical examples for MOFs, which are structurally rigid and do not allow for large volume changes (i.e. breathing transitions) under preservation of crystalline order. In our opinion, the occurrence of frustrated flexibility is merely a question of balancing the intra-framework interactions by the appropriate linker functionalization. Thus, the phenomenon of frustrated flexibility can principally arise in any structurally rigid framework compound, if the energetics of the framework are properly manipulated via linker functionalization with dispersion energy donors.

We have previously selected UiO-66/67/68, MIL-100/101 and HKUST-1 as other prototypical examples for MOFs with a structurally rigid framework. Similarly to MOF-5, appropriate functionalization might result in a contraction of these frameworks under loss of long-range order. We notice that out of these examples, HKUST-1 and MIL-100 are particularly difficult to functionalize, due to the restrictions of the 1,3,5-benzenetricarboxylate building unit. MOFs of the UiO-66/67/68 series as well as MIL-101, however, show considerable tolerance towards linker functionalization (see for example DOI: acs.chemmater.5b04575 and DOI: 10.1021/ic4005328) and thus could show frustrated flexibility once the intra-framework interactions are modified in a similar manner as presented here for the case of MOF-5. The higher connectivity of the UiO-type frameworks naturally results in more rigid structures. However, their propensity to form defects (and thus inorganic building units of lower connectivity) might facilitate the appearance of frustrated flexibility.

We have revised the text in our manuscript accordingly and removed HKUST-1 and MIL-100 from the examples.

8. In Figure S10.1, it seems that the nitrogen isotherm drops at higher pressures for dry-MOF-5-C2. Is this correct? If so, what is the origin of this drop?

Response: Most likely, this is caused by (i) an inaccurate dead volume measurement for this particular sample or (ii) a small leak in the measurement cell (microcrack in the glass tube or faulty sealing ring). However, the nitrogen sorption data are only provided for the sake of completeness. The data are irrelevant for the scientific conclusions drawn in our manuscript. We provided a more detailed discussion in our response to the comment on Figure S10.1 (called Supplementary Figure 9.1 in the revised version) of Reviewer #2 below.

9. For the temperature ramp simulations discussed in section S12, it seems the total simulation time and the ramp conditions are missing.

Response: We have added a sentence in the respective paragraph (Temperature ramp NPT simulations) stating these conditions (Section S12 is now called Supplementary Methods 11).

10. Are the differences in $p(V)$ curves when using initial states obtained either from the pressure ramp or the temperature ramp simulations connected with the range of $p(V)$ curves that can be obtained when having large sidechain functionalization (see for instance 10.1002/anie.202011004), or is this a completely separate effect?

Response: An excellent question allowing us to clarify and discuss possible effects of ‘the same’ system featuring distinct $p(V)$ profiles:

Considering the manuscript mentioned above, therein the spread in $p(V)$ curves originates from the choice of the initial structural models. Due to the absence of single crystal reference data, many distinct configurations were constructed, and these distinct configurations were shown to have different $p(V)$ profiles. We first encountered this behavior during this investigation (10.1039/D0FD00017E) for a similar system, $Zn_2(\text{BME-bdc})_2(\text{dabco})$, with BME-bdc = 2,5-bis(2-methoxyethoxy)-1,4-benzenedicarboxylate and dabco = 1,4-diazabicyclo[2.2.2.]octane.

Herein, we are using a systematic methodology to construct initial models based on the single crystal structure of *as*-MOF-5-C8, thereby ruling out the need to come up with structural models in the first place. Thus, we ensure that the structures are as comparable as possible, however, at the risk of potentially missing on other low-energy conformers, which may be important. However, a similar investigation as in DOI: 10.1002/anie.202011004 was not possible due to the mere numerical effort here. In the manuscript where the $Zn_2(\text{BME-bdc})_2(\text{dabco})$ system is discussed, we also ran multiple simulations for the very same initial configuration only differing in initial velocities, and found small deviations for the resulting profiles as well, which are a little smaller, but comparable to the deviations here.

We see two possible explanations for the difference in $p(V)$ profiles here, whereby in practice also both problems could be present in combination:

1. The configurations of the systems have changed in such a way that the resulting closing is different. Possibly during the closing of the structures, the probability of a configuration change diminishes and the system thus is trapped at the state where the phase transition initiates, thereby measuring the pressure response of this particular state.
2. The realizations of the phase transformation are on a different cell parameter trajectory, such that a subsequent sampling of $p(V)$ given these cell parameters results in different pressure profiles. This can, however, only happen if the system of a trajectory A does not undergo a transition to the cell parameters of another trajectory B or vice versa during sampling of $p(V)$, thus ultimately leading to insufficient sampling. We feel that this is a general problem of this methodology. We would be grateful for any kind of advice to resolve this issue and are currently working on aiding this with a combination of other enhanced simulation methodologies to explore the phase space at a given V in greater detail.

11. Can the authors pinpoint the origin of the large deviations with respect to the fitting curve for MOF-5 in Figure S12.4?

Response: This relates to the points discussed above (10). In these regions the structures are already very dense and would probably have decomposed already. Since MOF-FF is a non-reactive force field, bond cleavage is not allowed. In these high-density regions, however, we are not sure about the validity of the force field, and thus have to assume that these structures are not described correctly. Due to the way initial structures for the NV($\sigma_a=0$)T simulations are chosen, a smaller volume is not necessarily taken from a later stage of the preceding pressure or temperature ramp simulation. In addition, the denser structures are more prone to not equilibrate the cell parameters well. However, here we are only interested in a qualitative picture and the energetics of the parent MOF-5 in the open form (Figure S12 is now called Supplementary figure 11.4).

12. In the caption of Figure S12.6, the authors explain the significance of yellow, magenta, and dark blue boxes. However, it seems no explanation is given of the meaning of the light blue boxes.

Response: The reviewer is right about this. We have added a sentence stating the meaning of the light blue boxes in the figure caption of Figure S12.6 (now called Supplementary figure 11.6).

Reviewer #2 (Remarks to the Author):

In „Frustrated flexibility in metal-organic frameworks” Pallach et al. demonstrate the deformation of “rigid” MOF-5 derivatives that contain alkoxy side-chains, referred to as dispersion energy donors (DEDs) attached to the ligand backbone. The work is based on extensive experimental and computational studies and heavily builds on previous work of the authors about entropy-controlled deformation in flexible MOFs. It addresses the interesting situation in which internal dispersion interactions overcome the energy barrier for structural deformation of the framework lattice. In this work the authors utilize the MOF-5 framework with pcu topology which is in particular interesting as it is widely perceived as a rigid and non-deformable framework (topology).

I consider the paper a very interesting piece of work which I enjoyed reading. It certainly represents an important contribution to the field of flexible MOFs / soft porous crystals. The paper is well written and illustrated, uses appropriate citations and I highly recommend publication in Nature Communications considering the following suggestions which are of curious nature:

P3 I17 – with respect to crystalline – amorphous-crystalline transitions I would like to highlight recent work on mesoporous flexible MOFs (<https://pubs.rsc.org/en/content/articlelanding/2020/sc/d0sc03727c>). It seems that with extended framework backbone structural contraction occurs under enhanced loss of long-range order. The authors are free to cite the work or not. Current citations already seem appropriate.

Response: Indeed, this new article is relevant for our study. We cited the paper as ref. 32 in our revised manuscript.

P4 I1 – What about strong covalent bonds of the ligand itself? Hinging and buckling deformations of the ligand itself is an additional degree of freedom that can contribute to the deformation of the whole lattice? (this might actually become more relevant when the ligands are elongated).

Response: It is right that the covalent bonds of the linker are an important and very strong force within a MOF. Hinging and buckling deformations of the organic ligand are of relevance for some systems (such as DUT-49). We feel, however, that in the particular case of the MOF-5 derivatives discussed here, linker buckling and hinging are unimportant, due to the relatively small size of the bdc linkers. In the IRMOF-10-CX and MOF-177-CX derivatives presented here as well, such effects might nevertheless be of relevance. However, it is difficult to discuss this on the bases of the PXRD and IR data we have for these enlarged derivatives.

Nevertheless, we have included the “strong covalent bonds” as an important aspect for the energetic balance within a MOF in the sentence referred to by the reviewer.

P4 I4 – “distorted MOF phases exhibit the same topology” – I am no expert in the field and definition of topologies but can a distorted lattice be named topology? I would just state “same connectivity and bonding pattern”.

Response: The *underlying topology* or *net* of a crystalline material actually is defined solely by the “same connectivity and bonding pattern” (see e.g. DOI: 10.1021/cr200205j). Topologies are graphs, in particular labeled quotient graphs, whose vertices do not feature any position information in Euclidean

(3D) space. Removing position information is one of the steps during a topological deconstruction. The reverse process, in which the position information for a topology is restored is called embedding. For the present case (and generally for all flexible MOFs which do not undergo bond breaking and reformation during phase transitions) this means that the topology will stay the same no matter how much a structure is distorted, as long as the underlying connectivity is retained. Thus, our above statement is totally valid. In order to make this clear for the non-expert, we had already added “same connectivity and bonding pattern” to the statement in the previous version of our manuscript.

P4 I22 – “At first, we select an intrinsically rigid and non-responsive MOF structure type, which does not allow for correlated (and thus crystalline-to-crystalline) large-magnitude structural changes.” – Now this to me is a difficult statement because it lacks a clear definition of a large magnitude change. Staying with MOF-5 there are know variants of this framework (<https://pubs.acs.org/doi/abs/10.1021/jacs.5b11150>) that are structurally/symmetrically different that the original cubic phase. So maybe the authors can either differentiate their definition of large-magnitude against previous examples of deformations in the MOF-5 system, or at best provide a quantitative figure (let’s say for example 5% reduction in unit cell volume). \diamond I now see that the authors extensively quote on that later in the manuscript, still a quantification (like you provide on page 7 18) of “large” would be nice.

Response: We agree to this comment and have adjusted the addressed parts in the main text (page 4 and 5) by referring to the maximum volume change known from MOF-5 to undergo upon chiral induction (which is a volume reduction by about 4%).

P4 I26 – “thus change the energy landscape of the MOF structure.” – here I would clearly define which parameters define the energy landscape. In this case looking at the performed computation energy vs. unit cell volume and hydrostatic pressure.

Response: We revised the text and defined the important contributions to the potential energy landscape here (enthalpic and entropic contributions). The results of the computational and experimental work, which demonstrate these different contributions, are discussed in great detail in the corresponding sections later in the manuscript.

P4 I28 – with respect to frustration is there any kind of kinetic idea/concept associated to this state?

Response: Generally, the concept of frustrated flexibility was designed from a purely thermodynamic viewpoint. But the reviewer is right that kinetic effects might play a decisive role as well (as for many solid-state phase transformations, such as the hysteresis observed in sorption isotherms of breathing MOFs etc.). We think that the crystalline and the non-crystalline contracted phases of MOF-5-CX represent thermodynamic ground states of these compounds (degenerate ground states for the non-crystalline systems though), rather than kinetically trapped metastable intermediates. This is underlined by the DSC data, which demonstrate that the rhombohedrally distorted phases (in case of MOF-5-C7 and MOF-5-C8) are enthalpically favoured over the cubic phases. The non-crystalline phases of MOF-5-C2 to MOF-5-C6 are also enthalpically favoured compared to the cubic parent phase, if devoid of guest molecules.

P5 I24 – “since they would require major distortions of its inorganic nodes.” I would add bond breaking here which in a coordination cluster might also occur reversible (e.g. transmetalated MOF-5 derivatives by Dinca et al.) – I would also cite Ferey et al. here (<https://pubs.rsc.org/en/content/articlelanding/2009/cs/b804302g#!divAbstract>) as they were really the first to make that comparison

Response: Thanks for this suggestion. We have revised the text accordingly.

P10 I24 – “Remarkably, the mosaicity of the crystals does not increase after the guest-mediated crystalline-to-amorphous-to-crystalline transition. This indicates that the loss of crystallinity in dry-MOF-5-CX occurs mainly through a strain-based process, rather than cracking or bond breaking.”
◇ Very elegant!

With respect to the recovery of crystallinity do the authors have any idea to what extent the guest species contribute to the scattering? Unfortunately, there is no information provided in the ESI or manuscript if e.g. a squeeze method was performed after SCXRD refinement.

Response: As we have previously mentioned in the SI under Supplementary Methods 3.1, we have used the “solvent mask routine” implemented in OLEX2, which is very similar to the SQUEEZE routine of PLATON, for the treatment of the electron density of disordered CX side chains and solvent molecules. The full results of this treatment are included in the crystallographic information files already deposited at the CSD. We have now also included the number of masked (“squeezed”) electrons in the crystallographic tables of the SI (Supplementary Table 3.1-3.3).

We agree with the reviewer, that approaching the number of electrons covered by the solvent masking routine might be a way to assess contributions of guest molecules to Bragg scattering. Unfortunately, the dominance of Zn contributions to low angle diffraction intensity often leads to overexposure and therefore hampers the extraction of solvent contributions, as the masking routines mostly rely on the low angle reflection intensities. In the case of *oct*-MOF-5-C6 the crystals become twinned upon change of the solvent, which disables the solvent masking function.

In the other cases, except for the crystal structure of *as*-MOF-5-C8, only the O atom of the CX side chains could be resolved in the electron density map. Thus, the solvent mask also masks electrons stemming from the disordered CX side chains. In general, the number of masked electrons is in all structures much lower than the number expected from the size of the CX side chains and the amount of solvent guests per formula unit (as determined by NMR). For some structures the number of masked electrons is even zero. This suggests that a large fraction of the CX side chains, as well as the majority of the guest molecules, are so drastically disordered, they barely contribute to the Bragg scattering. Rather these groups give rise to diffuse scattering which is not taken into account in single crystal X-ray diffraction.

P13 I6 – “2 x 2 x 2 cavities in dry-MOF-5-C6.” – does that correspond to 2 x 2 x 2 unit cells? Maybe wise to use crystallographic description for a better correlation with the crystallographic results?

Response: This is a very good point. We have not made this clear enough in the previous version of the manuscript. Due to the presence of different phases and therefore different crystallographic unit cell settings within the MOF-5-CX series, the number of repeating units of the MOF contained in the

crystallographic unit cell varies (most structures feature one $Zn_4O(CX-bdc)_3$ formula unit per unit cell, others feature three or eight $Zn_4O(CX-bdc)_3$ formula units per crystallographic unit cell). For consistency, we changed the wording in the respective text passage from “cavity” to “cell” and define the term “cell” as the volume of one $Zn_4O(CX-bdc)_3$ repeating unit, to which we will also refer later in the main text. Each time simulation boxes are mentioned, supercells are given in reference to a single cell as defined previously (i.e. a supercell of 2 x 2 x 2 cells or a supercell of 8 x 8 x 8 cells).

P14 I18 – “In the smaller 2x2x2 simulation cells” – I assume this now correlates to unit cells?

Response: See above.

P14 I24 – I think at this point it is important to mention that the simulations were carried out on guest-free crystal structures. Maybe apply your dry-terminology as previously in the text. I think this is an important fact because in the real-world system the dry system is obtained by solvent removal and not directly made in the solvent-free state. In particular with respect to the presence and population of metastable states on the energy surface this is crucial.

Response: The referee is absolutely right about the importance of the absence of solvent in the simulations. However, we have mentioned this at the very beginning of the simulation section in both, the main text as well as the Supplementary Information. Furthermore, all initial structures were derived from the experimental *as*-MOF-5-C8 crystal structure, which naturally includes the synthetic history and is solvated (even though, the structure of MOF-5-C8 does not differ very much in the *as* and *dry* states). The referee is right in the sense as that we cannot fully exclude the possibility that we are working with metastable structures, and that simulation-wise, other structures would also be relevant. In particular, because the shorter side-chain MOFs were constructed from the C8 material, which does not necessarily have to be the most stable configuration.

Fig 4. – Out of interest: Has the energy landscape of MOF-5 without DED been calculated? Could be a good reference to probe the functionality of the force field.

Also maybe try to reference the 0 energy for a and b the same way being the energy minimum for the expanded large volume state. This way it might be easier to see the degree of stabilization for free vs. internal energy. Just a suggestion for better visualization.

Response: The parent MOF-5 systems’ energetics are given in the Supplementary Figure 11.4. Furthermore, we have added a small section where we report an energy scan of the parent MOF material during a rhombohedral distortion (see also our response to reviewer #1 comment 1 and 11). Moreover, we have tested the visualization suggestion and indeed found this representation to be the better one. Accordingly, we have changed Figure 4 to contain this version of the plot. We thank the referee for this suggestion.

Note that the differences in the relative internal energy ΔU between closed and open pore form for the different chain lengths, which becomes more visible in this form of the plot, is due to a different balance between entropy and enthalpy, presumably because of incomplete sampling. We note that we only discuss qualitative trends here.

The ability to match X-ray scattering with the simulated distorted structures is fantastic! Congrats.

P17 I12 – “The dispersion interactions introduced by the DEDs are apparently strong enough to outbalance the energetic cost for the deformation of the nodes” – I agree. Is it possible to derive the energetic costs from the MD simulations? For example which bond length elongation or dihedral deformation are the energetically most demanding distortions? – Can be material for another paper and I don’t consider it crucial for this particular study but certainly of interest in particular since the calculations have been performed.

Response: As already discussed in response to point 1 of reviewer #1 it is difficult to decompose the overall energetics into contributions due to the DEDs. We have made an estimate of the energy costs for the deformation and have added this in the Supplementary Information as “Strain energetics of the parent MOF-5 material”. Regarding the structural analysis, there is indeed potential for a deeper analysis. We have already discussed some issues in the “Average $O_{\text{eth}}-C_{\text{Methyl}}$ Distances” paragraph (see Supplementary Methods 11). However, due to the delicate balance between entropy and enthalpy in these systems, including both the framework and the flexible side chains, we believe it will not be possible to pinpoint a geometric origin for the observed energetic effects.

P18 I19 – “The thermal behaviour of dry-MOF-5-C6 is counterintuitive and inverse to the thermal behaviour of conventional materials.” – I am wondering if there are any comparable cases to this kind of behavior? I consider it quite outstanding, in particular given the stark contrast between MOF-5-C6 and MOF-5-C7

Response: We are not aware of any materials showing a behaviour similar to MOF-5-C6. See also our comment to point 2 of reviewer #1.

P20 I4 – “whereas all other derivatives only show N_2 adsorption on the external surface” I recommend using values of MOF-5 as a reference (can be published values) to illustrate the degree of pore occupation by DED. – As you do later for CO_2 at 195 K

Response: The N_2 capacity of all MOF-5-CX derivatives is strongly reduced (or basically zero for $X > 3$) due to the presence of the alkoxy groups (see Supplementary Figure 9.1 and 9.2). We assume that N_2 cannot effectively penetrate into the pores, if the alkoxy chains become longer than for MOF-5-C3. The situation is very different with CO_2 . Due to its smaller kinetic diameter and the higher temperature for data collection, CO_2 is readily adsorbed in the micropores of all MOF-5-CX compounds. Thus, the CO_2 isotherms provide a robust basis for the calculation of the average framework contraction, while the N_2 sorption data do not. In our opinion a calculation of the pore occupation by the DEDs based on the N_2 sorption data is meaningless because of the strong kinetic hindrance of N_2 sorption at 77 K.

Figure S10.1 – black isotherm exhibits a negative slope which I assume is caused by inaccurate dead volume measurement. I suggest repeating the experiment or at least commenting on this! I can

understand that the quantities required to accurately measure nitrogen isotherms for materials with such low porosity are difficult to achieve on this class of solids. A statement in the ESI will put the isotherms in perspective and avoid conflicts with the adsorption community.

Response: We agree here and added a detailed comment to Supplementary Figure 9.1, stating that this particular isotherm contains an error likely caused by (i) incorrect dead volume measurement or (ii) a small leak in the measurement cell (microcrack in the glass cell or a faulty sealing ring). The negative slope in the isotherm accounts for a drop in adsorbed volume of about 5% of the total volume adsorbed. The N₂ sorption isotherms are of minor importance for the present study, since neither quantitative nor qualitative information on the phenomenon of frustrated flexibility are extracted from these data.

Fig. 5b – Why would the authors expect a linear correlation? This is not clear to me neither from the figure nor the text. Also I could not find detail. Also be advised that a 1.5% change in unit cell volume can likely

I find the correlation/analysis of pore volume and cell contraction somewhat suitable but I am not sure if for example inaccessibility of pore space due to the DED could manipulate the adsorption experiments to a degree not detectable. The observed correlation might indicate that this is not the case, however I am unsure which “undetected” contributions could also be present.

Response: Without a doubt, there might be so far undetected contributions that can influence the sorption capacity of a rather polar adsorptive such as CO₂. Hence, we have to stress here that our approach needs to be considered as an estimation (as stated in the main text and in the corresponding detailed description in the SI). Importantly, the volumes of the contracted phases estimated from the sorption capacities are in good accordance with the volumes we estimate on the basis of the PXRD data of these compounds. This is a strong justification for this approach.

P20/123 – I suggest giving the respective temperatures (p20123) to the adsorption probes, not just in the first sentence but also further down in the text in particular for butane where temperature between adsorption and PXRD experiment varies. We have recently demonstrated the effect of temperature in DUT-49 (<https://pubs.rsc.org/en/content/articlelanding/2021/fd/d0fd00013b>) and believe these factors will be valid for most flexible or even frustrated MOFs. The statement “due to the stronger interaction of this gas with the hydrophobically lined interior of the pores 24 (polarizability $\alpha(\text{n-butane}) = 8.02 \text{ \AA}^3$ compared to $\alpha(\text{CO}_2) = 2.51 \text{ \AA}^3$)” is thus only partially accurate as the solid-fluid interactions are also a function of temperature relative for each respective guest.

Response: We agree to your objections and upgraded the relevant part of the main text.

P21 11 – Is it only a shift towards higher absolute pressure or also a shift to higher relative pressures. The latter would indicate a change in thermodynamics of the solid-fluid interactions, while a shift in absolute pressure is to be expected due to the change in temperature.

Response: The shift is also towards higher relative pressures. For further discussion, see below.

P21/I2 – “due to too short equilibration times” – is there any experimental support for this claim and what are the equilibration times?

Response: For the particular 50/100 kPa measuring points of the *in situ* PXRD during *n*-butane sorption experiment of *dry*-MOF-C7 only equilibration times of about 2 min were applied, whereas the equilibration times during the pure sorption experiments were 10 min. Also, during the *in situ* PXRD experiments the density of measuring points around the transition pressure is significantly lower (2 points in the *in-situ* PXRD experiment vs. 22 points in the pure sorption experiment). We suggest, the presence of the pore-filling DEDs slows down the kinetics of the sorption process, as it heavily limits the ability of the guest molecules to diffuse into the pores of the material. Thus, the equilibration time should have a major influence on the observed deviation of the relative transition pressure regime rather than the temperature (in this particular case here), since it is only deviating by $\Delta T = 5$ °C between the *in situ* PXRD and the pure *n*-butane sorption experiments (p_0 changes from 2.1 bar to 2.4 bar when increasing the temperature from 293 to 298 K).

The equilibration times were added to the corresponding paragraph in the Methods section.

P21/I4 – “the non-crystalline distorted phase to the crystalline cubic phase with increasing *n*-7 butane pressure.” Previously in the paper the authors demonstrate reversible transitions by suspension in *n*-octane. I suggest to correlate the observations of this suspension and provide timescale required for these transformations (e.g 2 days in octane vs. 10 min equilibration for *n*-butane adsorption)

Response: As stated before, within the scope of this study, we have mainly focused on investigating thermodynamic ground states, therefore only few data on the kinetics of the investigated structural transformations are available at the moment. The transformations in *n*-octane suspension were not investigated regarding their kinetics. The MOFs simply have been immersed in *n*-octane at 65 °C for 16 h before structural analysis. That means, the phase transitions upon *n*-octane adsorption may be accomplished anytime between “directly after addition of *n*-octane” or “just after 16 h in *n*-octane” at 65 °C. Thus, a comparison between the transformation kinetics upon *n*-butane and *n*-octane adsorption is not possible at the moment. We aim to look into the transformation kinetics of these systems via time dependent PXRD measurements in the future.

P21/I8 – “After pressure release, MOF-5-C6 does not immediately show retransformation to the non-crystalline phase.” Can you provide a time that you consider “not immediate”? Quantification would be very helpful as kinetic processes in framework deformation are currently intensely discussed.

Response: Here, “not immediately” refers to the timescales of the experiment, which is about 10 min, the equilibration time of the measuring points for *dry*-MOF-5-C6. To clarify this, we have added the return PXRD patterns that were collected for the respective material to the Supplementary Information (Supplementary Figure 10.9) and added a few words to the concerning part of the main text. In addition to the return PXRD pattern that was recorded directly after releasing the *n*-butane pressure, a second PXRD pattern collected 17 h after the release of *n*-butane pressure (sample stored under

ambient conditions) is also included. The last pattern shows that *dry*-MOF-5-C6 has largely returned to the initial non-crystalline phase at that point. Unfortunately, we cannot quantify the transformation kinetics on the basis of our current data. As mentioned in our response to the previous comment of this reviewer, we aim to look into the kinetics in the future.

Can the authors provide cif files of the MOF materials and the models used for computation as supplementary information?

Response: Regarding the simulations, there will soon be a github repository with initial structures, run scripts and force field parameters available for each individual simulation carried out.

https://github.com/cmc-rub/supporting_data/tree/master/90-Pallach-Keupp-NatCommun

We have added a paragraph “Data Availability” to the main text where the link is given.

NOTE: this repo is not public as of 06. May 2021. It will be made public once the paper is publication ready.

I want to express my support for submitting such an extensive study in a single paper! I think this work could have easily been spilt up in 2-3 individual papers but a lot would have been lost in that process! The paper really benefits from the large body of experimental and computational work! Congrats and well done managing such an extensive volume of work!

In 2015 at a Symposium on flexible MOFs in Paris (to which some of the authors attended as well) Gérard Férey insisted after a lively discussion on flexibility that MOF-5 cannot be flexible. The proposed concept of “frustrated flexibility” described in this work illustrates that we have a long way to go in knowing what can and what cannot be “flexible”. Well done!

Lastly, a comment:

While I do in general agree to the term of frustrated flexibility in the case described here, I would be careful applying it/extending it to all kinds of responsive flexibility in framework materials. I would associate frustration in the given context to some kind of kinetically trapped state which to some extent is present in the investigated system. In the future I would hope for a more concise and extended definition of such framework frustration. In essence, any framework material can be compressed in a denser state by application of hydrostatic pressure. Whether this can occur reversibly mainly depends on the absence of framework disintegration. The uniqueness of the present system (in my opinion) arises from the temperature responsiveness. It also seems that the presence of DED overcomes a lot of issues associated with framework disintegration and furthermore overcomes issues associated with guest induced transitions (e.g. diffusion, concentration gradients, accessibilities, etc.). They seem to be a perfect system to systematically study entropic effects.

Dr Simon Krause

REVIEWERS' COMMENTS

Reviewer #1 (Remarks to the Author):

The authors have extensively commented on the points raised during the review process. I'm therefore very happy to suggest the revised version of this interesting manuscript for publication in Nature Communications, and I laud the authors for their thorough work presented here.

Reviewer #2 (Remarks to the Author):

I would like to thank the authors for providing extensive revisions and addressing all questions raised. I am happy to see this paper published.

RESPONSE TO REVIEWER COMMENTS

Reviewer #1 (Remarks to the Author):

The authors have extensively commented on the points raised during the review process. I'm therefore very happy to suggest the revised version of this interesting manuscript for publication in Nature Communications, and I laud the authors for their thorough work presented here.

Response: We are delighted that the reviewer is happy with our revisions and is suggesting publication of our work in Nature Communications.

Reviewer #2 (Remarks to the Author):

I would like to thank the authors for providing extensive revisions and addressing all questions raised. I am happy to see this paper published.

Response: We thank the reviewer for the very positive response.